# Peripheral-specific Y1 receptor antagonism increases thermogenesis and protects against diet-induced obesity

Chenxu Yan[1,2,10], Tianshu Zeng[3,10], Kailun Lee [1,10], Max Nobis[4,5], Kim Loh[1,8], Luoning Gou[3], Zefeng Xia[3], Zhongmin Gao[2], Mohammed Bensellam [2,9], Will Hughes[2,5], Jackie Lau[1], Lei Zhang [1,5], Chi Kin Ip [1,5], Ronaldo Enriquez[1], Hanyu Gao[2], Qiao-Ping Wang[6], Qi Wu[2], Jody J. Haigh[7], D. Ross Laybutt[2,5], Paul Timpson[4,5], Herbert Herzog [1,5,11✉] & Yan-Chuan Shi [1,2,5,11✉]

Obesity is caused by an imbalance between food intake and energy expenditure (EE). Here we identify a conserved pathway that links signalling through peripheral Y1 receptors (Y1R) to the control of EE. Selective antagonism of peripheral Y1R, via the non-brain penetrable antagonist BIBO3304, leads to a significant reduction in body weight gain due to enhanced EE thereby reducing fat mass. Specifically thermogenesis in brown adipose tissue (BAT) due to elevated UCP1 is enhanced accompanied by extensive browning of white adipose tissue both in mice and humans. Importantly, selective ablation of Y1R from adipocytes protects against diet-induced obesity. Furthermore, peripheral specific Y1R antagonism also improves glucose homeostasis mainly driven by dynamic changes in Akt activity in BAT. Together, these data suggest that selective peripheral only Y1R antagonism via BIBO3304, or a functional analogue, could be developed as a safer and more effective treatment option to mitigate diet-induced obesity.

[1] Neuroscience Division, Garvan Institute of Medical Research, St Vincent's Hospital, Sydney, NSW, Australia. [2] Diabetes and Metabolism Division, Garvan Institute of Medical Research, St Vincent's Hospital, Sydney, NSW, Australia. [3] Wuhan Union Hospital, Huazhong University of Science and Technology, Wuhan, Hubei, China. [4] Invasion and Metastasis Lab, Cancer Division, Garvan Institute of Medical Research, St Vincent's Hospital, Sydney, NSW, Australia. [5] Faculty of Medicine, UNSW Australia, Sydney, NSW, Australia. [6] School of Pharmaceutical Sciences (Shenzhen), Sun Yat-sen University, Guangzhou, China. [7] Research Institute in Oncology and Hematology, Department of Pharmacology and Therapeutics, University of Manitoba, Winnipeg, MB, Canada. [8] Present address: St. Vincent's Institute of Medical Research, Fitzroy, VIC, Australia. [9] Present address: Institute of Experimental and Clinical Research, Pole of Endocrinology, Diabetes and Nutrition, Université catholique de Louvain, Brussels, Belgium. [10] These authors contributed equally: Chenxu Yan, Tianshu Zeng, Kailun Lee. [12] These authors jointly supervised this work: Herbert Herzog, Yan-Chuan Shi. ✉email: h.herzog@garvan.org.au; y.shi@garvan.org.au

Obesity and metabolic disease occur when energy intake consistently exceeds energy expenditure (EE). Current available therapies targeting modulation of food intake (FI), with the exception of bariatric surgery, have not proven effective in reducing and maintaining fat mass and body weight. One of the main reasons for this is that therapies targeting central appetite control also influences a variety of anxiety-related behaviours and emotionality[1,2] that can lead to severe side effects that diminish the efficacy of such interventions.

One of the most powerful controllers of feeding and energy homeostasis regulation is neuropeptide Y (NPY), which is widely expressed in the central nervous system exerting its actions through signalling via Y1, Y2, Y4, Y5 and y6 receptors[3]. Negative energy balance leads to an elevation of hypothalamic NPY levels triggering an increase in FI and a simultaneous decrease in EE mostly by inhibiting sympathetic output, leading to decreased brown adipose tissue (BAT) thermogenesis as well as browning of white adipose tissue (WAT)[4,5]. Together this highlights the critical direct role of central NPY controlling thermogenic activities of BAT and WAT. However, central NPY signalling also influences a variety of other physiological processes that are linked to altering mood and anxiety[6,7] thereby limiting its potential use as a clinically feasible target for appetite and EE intervention.

Importantly, however, NPY and its Y-receptors are also prominently expressed in the periphery where NPY is co-stored and co-released with noradrenaline from postganglionic sympathetic neurons and chromaffin cells of the adrenal medulla[8]. In addition, peripheral tissues and cells such as adipose tissue, pancreas, liver, skeletal muscle, and osteoblasts have been shown to express NPY[9]. While the pharmacologically similar acting other members of the NPY family, peptide YY (PYY) and pancreatic polypeptide (PP), can also be found in the pancreas and the gut[10]. More importantly, the corresponding Y-receptors are found to be expressed in key peripheral metabolic organs, including adipose tissue and pancreas, indicating a potential direct function of the NPY system in the local control of glucose and energy homeostasis. However, the nature and mechanisms whereby Y-receptor signalling that modulate metabolism and energy homeostasis in the periphery are largely unknown. The few exceptions are the recent identification of the inhibitory tone that the Y1R signalling has on insulin secretion in pancreatic β cells[11] and the beneficial effects that partial knockdown of Y1R signalling inhibition in peripheral tissues has on fat mass and lipid oxidation[9]. These results lead us to hypothsize that blocking Y1R signalling in peripheral tissues may ameliorate diet-induced obesity (DIO) and improve whole-body glucose metabolism.

To test this, we used several in vivo models for selective Y1R antagonism and also examined its effects ex vivo in primary human adipose tissues. Specifically, we employed the peripheral only acting Y1R selective antagonist BIBO3304[12] to avoid any interference with central Y1R signalling pathways and investigated its effects via comprehensive metabolic profiling especially focusing on the pathways that influence EE in various mouse models.

## Results

**Elevated Y1R expression under obese conditions in mice and humans.** Genetic association studies have linked the Leu7/Pro polymorphism of the NPY gene to elevated circulating NPY level and the development of obesity[13–15]. Therefore, dampening the NPY-ergic tone could be an effective approach for weight loss. In order to test our hypothesis of NPY signalling in the periphery being critical in the control of energy homeostasis and to identify the most likely Y-receptor(s) mediating this effect, we fed wild-type (WT) C57Bl/6JAusb mice a high-fat diet (HFD) for 7 weeks

and then examined Y-receptor mRNA expression levels in a variety of metabolically important tissues including the inguinal white adipose tissue (WATi) (representing subcutaneous fat), the epididymal WATe (representing intra-abdominal fat), BAT, liver, and skeletal muscle. Importantly, Y1R mRNA expression in BAT and WATi, but not in WATe was significantly elevated after exposure to HFD feeding (Fig. 1a). Y1R expression in response to HFD feeding was also upregulated in muscle, but was not altered in liver tissue, which displayed very low basal Y1R mRNA levels (Supplementary Fig. 1a). Under chow feeding conditions, Y2R expression was absent in BAT, but detectable in WATi and WATe (Supplementary Fig. 1b). In response to HFD, Y2R expression did not change in WATi, but was reduced in WATe (Supplementary Fig. 1b). Y4R expression was detected in WATe and BAT, and was increased by HFD in WATe but not in BAT (Supplementary Fig. 1c). Y5R expression was detected in all three fat depots, and was significantly upregulated with HFD feeding in WATi, but was not altered in BAT and WATe (Supplementary Fig. 1d). The high levels of Y1R expression in fat depots compared to much lower basal expression levels of Y2R, Y4R and Y5R suggests that signalling through Y1R may be the major contributing factor to NPY-driven energy controlling processes.

In order to determine whether Y1R expression is altered under conditions of obesity in humans, subcutaneous adipose tissue (SAT) and visceral adipose tissue (VAT) biopsies were collected from adult subjects with BMI classified as normal (BMI < 24.9) or obese (BMI > 30), and the expression pattern of Y-receptor mRNAs were determined. Consistent with the upregulation of Y1R expression in adipose tissue from mice, Y1R expression in SAT and VAT fat depots from obese human subjects was markedly elevated compared to that from lean subjects (Fig. 1b, c). In comparison, the basal expression levels of Y2R, Y4R were very low in both SAT and VAT of lean and obese subjects, while Y5R expression levels were slightly upregulated in VAT but not in SAT of obese subjects (Fig. 1b, c), suggesting that, similar to the mouse, in humans Y2R, Y4R and Y5R are unlikely to play a major role in NPY sigalling in adipose tissue. These data highlight a conserved role of Y1R signalling and the development of obesity across species. Interestingly, our analysis also revealed a positive correlation between Y1R expression in SAT depot and BMI (Fig. 1d, Pearson correlation coefficient $r^2 = 0.198$, $p = 0.019$). Also no relationship between Y1R expression in the intra-abdominal VAT depot and BMI was found in either lean or obese subjects (Supplementary Fig. 1e, f), suggesting that upregulated expression of Y1R in subcutaneous fat is linked to the development of obesity. Taken together, Y1R expression is consistently upregulated in adipose tissues of both obese mice and humans, suggesting a functional link between Y1R signalling and the development of obesity.

**Selective Y1R antagonism in the periphery prevents DIO.** To functionally test whether peripheral Y1R signalling influences body weight gain and body composition under conditions of a positive energy balance, 8-week-old WT mice were fed either a chow or a HFD in the presence or absence of the non-brain-penetrable Y1R selective antagonist BIBO3304[12] over a 7-week monitoring period (between 8 and 15 weeks of age) (Fig. 1e). On a normal chow diet, body weight in BIBO3304-treated mice was not significantly different from that in vehicle-treated mice when expressed as absolute body weight (Fig. 1f), although there was a tendency for reduced body weight gain (Fig. 1g). Importantly, BIBO3304-treated mice gained significantly less body weight on a HFD compared to vehicle-treated control mice, both in absolute body weight (Fig. 1f) and body weight gain (Fig. 1g). The

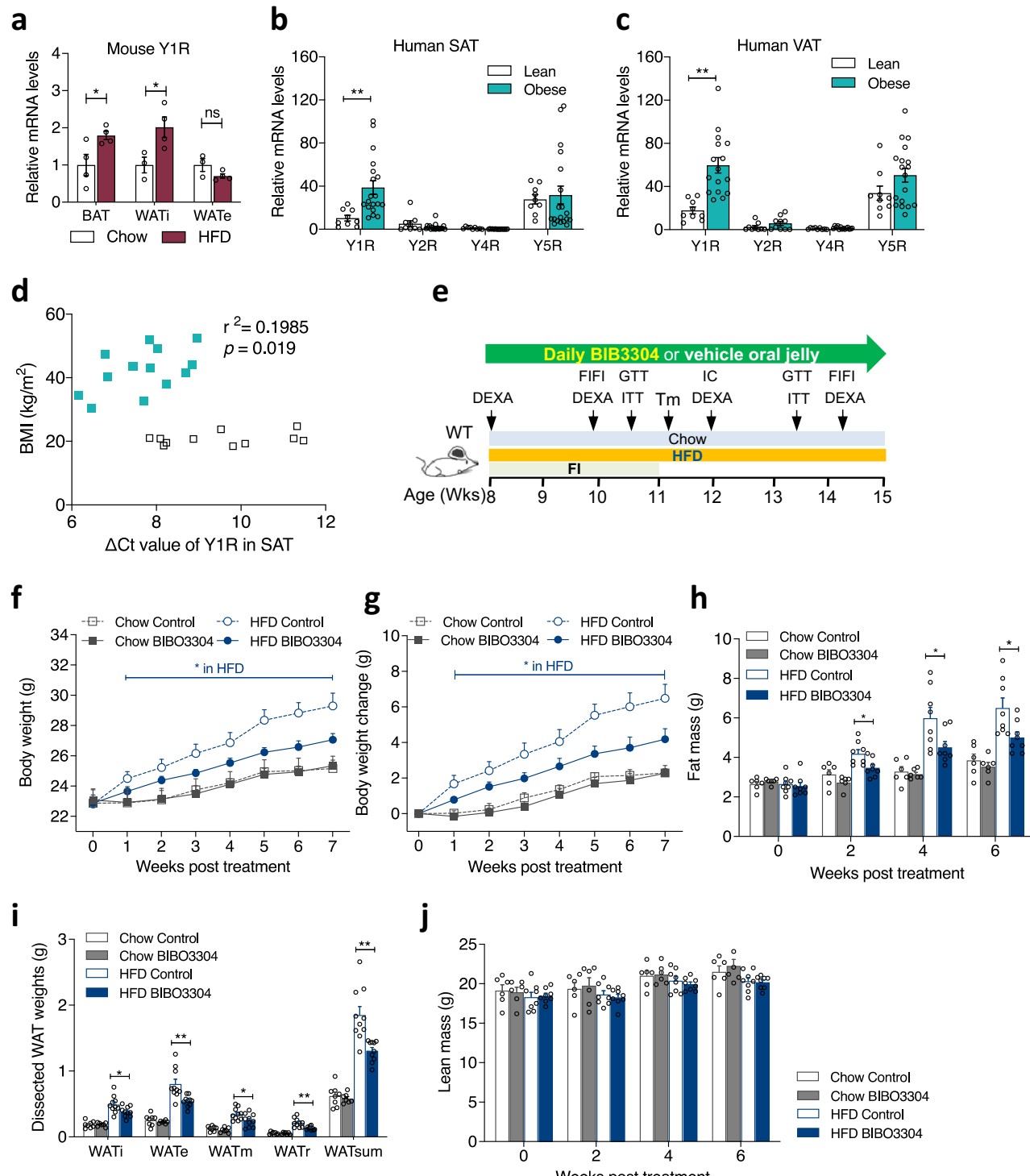

reduction in body weight gain in BIBO3304-treated DIO mice is entirely due to reduced fat mass accumulation during the treatment period as determined by DEXA at 4 and 6 weeks post treatment (Fig. 1h). The reduction in whole-body fat mass is further confirmed by a marked reduction in the weights of 4 major dissected white adipose depots (inguinal, epididimal, mesenteric and retroperitoneal), whether expressed in absolute weight (Fig. 1i) or as a percent of body weight (Supplementary Fig. 1g). In contrast, whole-body lean mass (Fig. 1j) determined by DEXA was not significantly altered by BIBO3304 treatment under either chow or high fat feeding.

**Periphery-selective Y1R antagonism increases EE.** To identify which side of the energy balance equation is altered by BIBO3304 treatment, we firstly measured spontaneous FI in both chow- and HFD- fed mice. The average daily food intake (Fig. 2a) and cumulative FI over the first 3 days of treatment did not differ (Supplementary Fig. 2a) and daily (Supplementary Fig. 2b) and cumulative FI measured from day 6 to day 21 (Fig. 2b), and at 5 weeks (Supplementary Fig. 2c) post treatment was indistinguishable between control and BIBO3304-treated mice on either chow or HFD feeding. Specifically, control and BIBO3304-treated mice showed similar spontaneous FI during the dark

**Fig. 1 Y1R mRNA expression in mice and human adipose tissue as well as effect of peripheral Y1R antagonism on body weight and body composition in wild-type mice. a** *Y1R* mRNA expression in brown adipose tissue (BAT, $n = 4$ per group), inguinal white adipose tissue (WATi, chow $n = 3$, HFD $n = 4$) and epididymal WAT (WATe, chow $n = 3$, HFD $n = 4$) in wild-type mice fed with a standard chow (white bar) or 7-week high fat diet (HFD, magenta bar). Data are mean ± s.e.m. *$p < 0.05$ by two-tailed $t$ test compared to the same adipose depot of chow-fed mice. ns non-significance. **b, c** Y-receptor mRNA expression profiles in subcutaneous white adipose tissue (SAT) and visceral white adipose tissue (VAT) from lean ($n = 9$, white bars) and obese ($n = 19$, cyan bars) non-diabetic human subjects relative to *Y4R* expression of lean subjects. Data are mean ± s.e.m. **$p < 0.01$ by two-tailed $t$ test compared to lean subjects. **d** Correlation between body mass index (BMI, kg/m$^2$) and ΔCt values of *Y1R* mRNA expression in SAT of obese (cyan filled square, $n = 16$) and lean (open square, $n = 11$) individuals. Data are mean ± s.e.m. $p$ values by two-tailed Pearson Correlation analysis. **e** Schematic illustration of phenotypic paradigm. WT wild type, DEXA Dual-energy X-ray absorptiometry, FI food intake, FIFI fasting-induced food intake, Tm temperature measurement by thermal camera, IC indirect calorimetry by Promethion (Sable Systems), GTT glucose tolerance test, ITT insulin tolerance test. **f, g** Absolute body weight and the change of body weight of wild-type mice on a chow or a HFD treated daily with a jelly containing either vehicle or Y1R-specific antagonist BIBO3304 for 7 weeks. Data are mean ± s.e.m. Chow $n = 6$ (grey, control: open circle; BIBO3304: filled circle), HFD control $n = 10$ (blue, open square), BIBO3304 $n = 11$ (filled square). *$p < 0.05$, two-way repeated measures ANOVA. **h** Whole-body fat mass of wild-type mice determined by DEXA at baseline, 2, 4 and 6 weeks after daily administration with control jelly or BIBO3304 jelly. Data are mean ± s.e.m. Chow $n = 6$ (control: open grey bar; BIBO3304: grey bar), HFD $n = 8$ (control: open blue bar; BIBO3304: blue bar). *$p < 0.05$: one-way ANOVA with Tukey's multiple comparisons test. **i** Dissected weights of individual WAT from inguinal (WATi), epididymal (WATe), mesenteric (WATm), retroperitoneal (WATr) and total weights of these 4 depots (WATsum) at cull. Data are mean ± s.e.m. Chow $n = 7-8$ (grey), HFD $n = 10$ (blue). *$p < 0.05$, **$p < 0.01$: one-way ANOVA with Tukey's multiple comparisons test. **j** Whole-body lean mass determined by DEXA. Data are mean ± s.e.m, Chow $n = 6$ (control: open grey bar; BIBO3304: grey bar), HFD $n = 8$ (control: open blue bar; BIBO3304: blue bar), $p$ values by one-way ANOVA with Tukey's multiple comparisons test. Source data are provided as a Source Data file.

phase, the light phase and the 24-h time period (Fig. 2c). There was also no significant difference in feeding efficiency between the treatment or diet groups (Supplementary Fig. 2d). Furthermore, similar to spontaneous FI under fed condition, no differences in 24-h fasting-induced FI was evident between BIBO3304-treated and vehicle-treated mice at week 2 (Supplementary Fig. 2e) and week 6 (Supplementary Fig. 2f) post treatment, fed either a chow or HFD. In response to 24-h fasting, the relative weight loss with respect to pre-fasting body weight and body weight recovery after refeeding was also similar between genotypes under both diets (Supplementary Fig. 2g, h). In addition, no differences in daily faecal output were observed (Supplementary Fig. 2i). Taken together, these data clearly rule out energy intake as the cause of reduced body weight gain seen in the BIBO3304-treated mice. Importantly, this lack of change in FI also demonstrates no involvement of central Y1R signalling in BIBO3304-induced body weight reduction, further confirming the peripheral selective action of BIBO3304.

We next examined whether BIBO3304 treatment affects EE. Under chow feeding, BIBO3304-treated mice showed similar EE to vehicle-treated mice (normalized to metabolically active tissue (MAT) weight[16,17]) (Fig. 2d, e). As expected, both HFD groups displayed an overall increase in EE compared to the chow group. Importantly, however, the BIBO3304-treated HFD group showed a significantly higher EE than the vehicle-treated HFD group (Fig. 2d, e), indicating that increased EE being a primary contributing factor to the reduced body weight gain and fat mass accretion in mice with peripheral selective Y1R antagonism. In addition, no change in physical activity was detected in any of the treatment groups (Supplementary Fig. 2j, k), excluding altered physical activity as a potential cause for the increase in EE.

Interestingly, respiratory exchange ratio (RER), an indicator of metabolic fuel preference, was significantly lower during the dark phase in chow-fed mice with BIBO3304 treatment, but this was reversed during the light phase (Fig. 2f, g), resulting in no overall difference in RER over the 24-h period (Fig. 2g). Under HFD conditions, RER was lower compared to chow conditions for both control and BIBO3304-treated mice (Fig. 2f), indicating a greater use of lipids as oxidative fuel with high-fat feeding. However, RER was significantly lower in HFD-fed BIBO3304-treated mice compared to vehicle-treated control mice in both, the light and the dark phases (Fig. 2f, g). These results suggest that BIBO3304 treatment causes a shift towards increased fat utilization

especially during the more active dark phase in both chow- and HFD-fed mice.

**Selective peripheral Y1R antagonism improves glucose tolerance.** To investigate whether Y1R antagonism is associated with metabolic benefits, we next determined insulin levels in mice treated with BIBO3304 in both chow and HFD groups under fed condition. Chow-fed BIBO3304-treated mice exhibited a tendency towards higher insulin levels compared to controls ($p = 0.08$, Fig. 3a) in the fed state, consistent with previous results showing that Y1R signalling inhibition in islets increases insulin secretion[11]. HFD-fed mice showed higher insulin levels than chow-fed mice as expected, but there were no differences observed between treatment groups under HFD conditions (Fig. 3a). Glucose tolerance tests (GTTs) were conducted at 2- and 6 weeks post BIBO3304 treatment, respectively. Under a chow diet, serum glucose (Fig. 3b) and insulin (Fig. 3c) levels during GTT showed no difference between the treatment groups at week 2. As expected, HFD-fed mice showed worsened glucose responses compared to the chow-fed mice (Fig. 3b) at week 2. However, mice treated with BIBO3304 showed improved glucose tolerance, which reached statistical significance at the 15- and 30-min time points. Interestingly, this was associated with unaltered insulin levels (Fig. 3c).

At 6 weeks post treatment, chow-fed BIBO3304-treated mice again displayed a similar glucose excursion during a GTT relative to vehicle-treated mice (Fig. 3d). In stark contrast, while HFD-fed vehicle-treated mice displayed marked glucose intolerance, this was almost completely normalized to chow-fed control levels in the BIBO3304-treated HFD mice (Fig. 3d). Again, the improvement in glucose tolerance was associated with unaltered insulin levels (Fig. 3e).

To evaluate insulin responsiveness in these mice, we next performed insulin tolerance tests (ITT). As expected, HFD-fed mice were less insulin responsive than chow-fed mice at 2 (Fig. 3f) and 6 weeks (Fig. 3g) post high fat feeding. Insulin responsiveness in BIBO3304-treated animals within the same diet was not different from vehicle-treated mice at the 2 week treatment time point (Fig. 3f). However, at 6 weeks post treatment, HFD-fed BIBO3304-treated mice showed improved insulin tolerance compared to vehicle-treated mice (Fig. 3g), displaying the same AUC value as chow-fed mice. Collectively, these results indicate that selective peripheral antagonism of Y1R

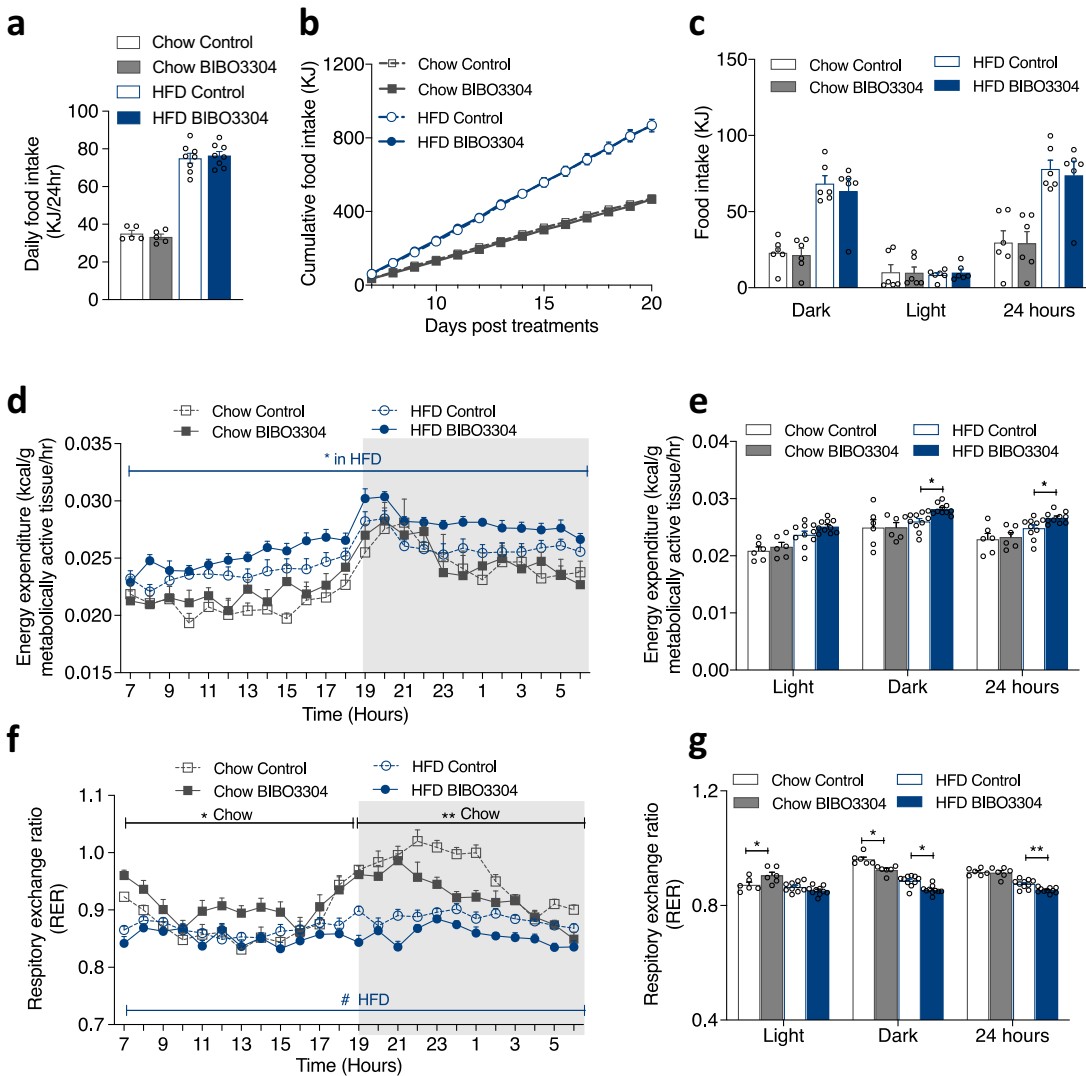

**Fig. 2 Effect of peripheral Y1R antagonism on food intake and energy expenditure in wild-type mice. a** Spontaneous daily food intake over first 3 days of wild-type mice treated daily with a jelly containing either vehicle or BIBO3304 on a chow or a HFD at 8 weeks of age, Data are mean ± s.e.m, Chow $n = 5$ (control: open grey bar; BIBO3304: grey bar), HFD $n = 8$ (control: open blue bar; BIBO3304: blue bar). $p$ values by one-way ANOVA. **b** Cumulative food intake of chow- and HFD-fed wild-type mice treated daily with a jelly containing either vehicle or BIBO3304 from day 6 to day 20 post treatment. Data are mean ± s.e.m, Chow $n = 5$ (grey, control: open square; BIBO3304: filled square), HFD $n = 8$ (blue, control: open circle; BIBO3304: filled circle). $p$ values by two-way repeated measures ANOVA. **c** Food intake during the dark phase, light phase and over 24 h period of chow- and HFD-fed wild-type mice treated daily with a jelly containing either vehicle or BIBO3304 at 8–9 weeks of age. Data are mean ± s.e.m, $n = 6$ per group. $p$ values by one-way ANOVA. **d** Energy expenditure (normalized to metabolically active tissues) over a 24-h course, with **e** showing average energy expenditure during the light phase, dark phase and over a 24-h period from (**d**). **f** Respiratory exchange ratio (RER) over a 24-h period, with (**g**) showing RER during the light phase, dark phase and over a 24-h period from (**f**). Data (**d**, **e**, **f**, **g**) are mean ± s.e.m. Chow $n = 6$, HFD $n = 10$, *$p < 0.05$, **$p < 0.01$ vs control in the same diet, one-way ANOVA with Tukey's multiple comparisons test. Source data are provided as a Source Data file.

leads to measurable improvement in glucose homeostasis under diet-induced obese conditions.

**Peripheral selective Y1R signalling controls body and BAT temperature**. A key aspect of EE is to alter body and BAT temperature to adjust energy homeostasis depending on energy surplus or deficit. To evaluate this, we employed a non-invasive infra-red camera detection system allowing us to measure skin surface temperature above the BAT as well as a region over the lumbar spine (back), which represents BAT and core body temperature, respectively (Fig. 4a). Body temperature measured around the lumbar spine region of BIBO3304-treated mice on both chow diet and HFD was significantly increased (Fig. 4b).

Similarly, the temperature of BAT, a primary thermogenic organ that is particularly critical for heat dissipation, was also significantly increased by BIBO3304 treatment under both chow and HFD conditions (Fig. 4c). Importantly, this increase was independent of BAT weight, which was unchanged by BIBO3304 treatment (Fig. 4d). We also assessed the difference between BAT and lumbar temperatures to examine whether BAT activation primarily accounts for driving thermogenesis. Importantly, the difference between the BAT and lumbar spine temperature significantly differed under HFD diet but not under chow diet conditions (Fig. 4e), indicating BAT thermogenesis is significantly contributing to the increase in whole-body EE. Furthermore, it is important to note that an increase in body temperature may not only be due to increased thermogenesis but could also be caused

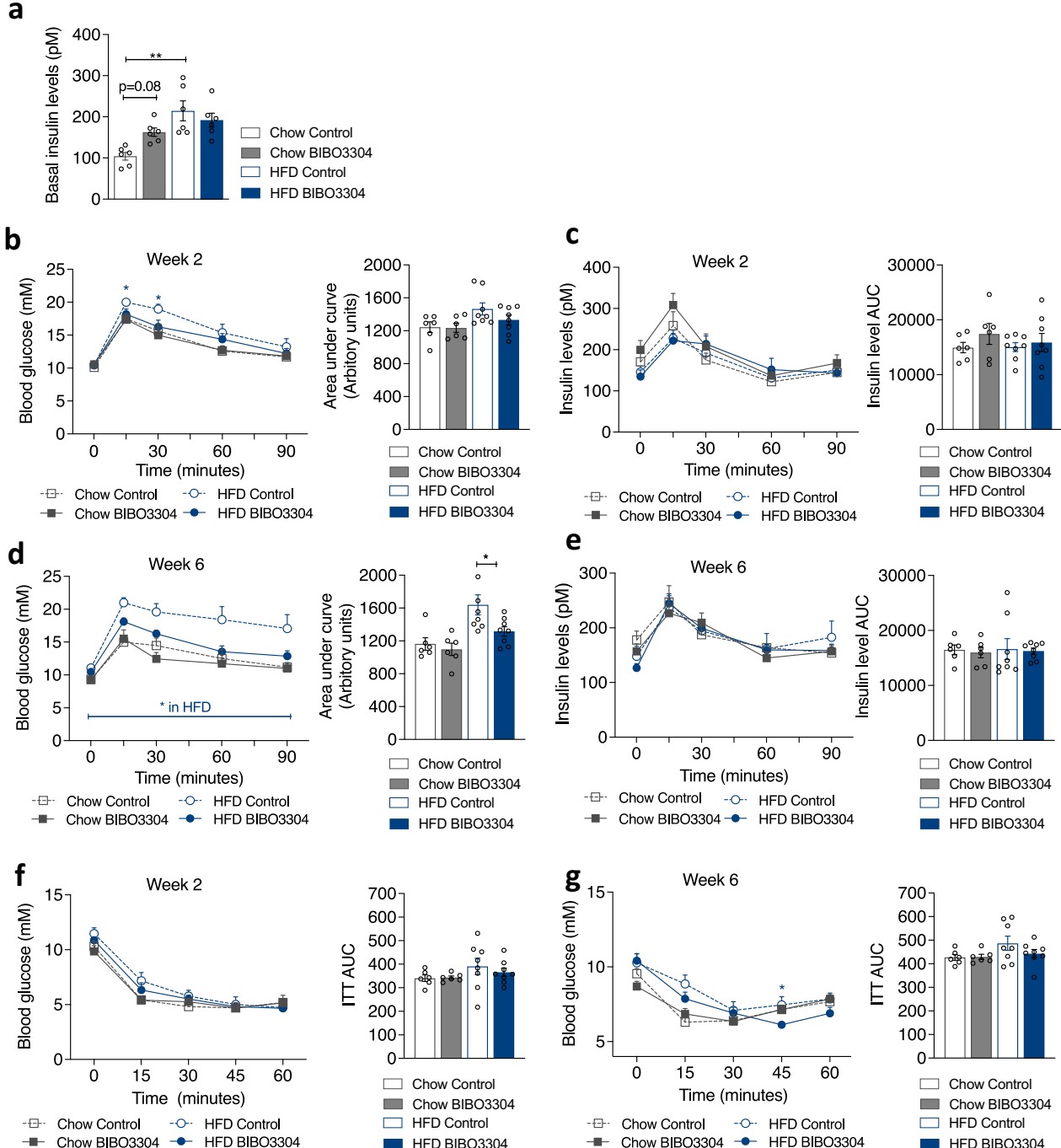

**Fig. 3 Effect of peripheral Y1R antagonism on glucose homeostasis. a** Blood insulin levels under fed condition in chow- or HFD-fed wild-type mice given either a chow or a HFD at 13–14 weeks of age. Data are mean ± s.e.m, $n = 6$ per group (chow, control: open grey bar; BIBO3304: grey bar. HFD, control: open blue bar; BIBO3304: filled blue bar). *$p < 0.05$, **$p < 0.01$, one-way ANOVA with Tukey's multiple comparisons test. Glucose levels in intraperitoneal glucose tolerance tests (GTT) in chow- or HFD-fed wild type mice after 2 (**b**) and 6 weeks (**d**) daily treatment with vehicle or BIBO3304, and corresponding area under the curve (AUC) analysis of glucose over time in these mice. **c, e**, Insulin levels throughout the GTTs 2 and 6 weeks post treatments in (**b, d**) and corresponding AUC analysis for these mice. **f, g**, Glucose levels in an insulin tolerance test (ITT) conducted after 2 and 6 weeks daily treatment with either vehicle or BIBO3304, and corresponding AUC analysis for the mice in (**f, g**). Data are mean ± s.e.m, chow $n = 6$ (control: open grey; BIBO3304: grey), HFD $n = 8$ (control: open blue; BIBO3304: blue), *$p < 0.05$, two-way repeated measures ANOVA with Sidak's multiple comparisons test. Source data are provided as a Source Data file.

by reduced heat loss. As the tail is regarded as a major thermo-regulatory organ in heat dissipation in rodents, and an increase in tail skin temperature is indicative of an increase in heat loss[18], tail

temperature was examined. Interestingly, in contrast to chow-fed mice, BIBO3304-treated DIO mice showed a significant increase in tail temperature compared to control DIO mice (Fig. 4f),

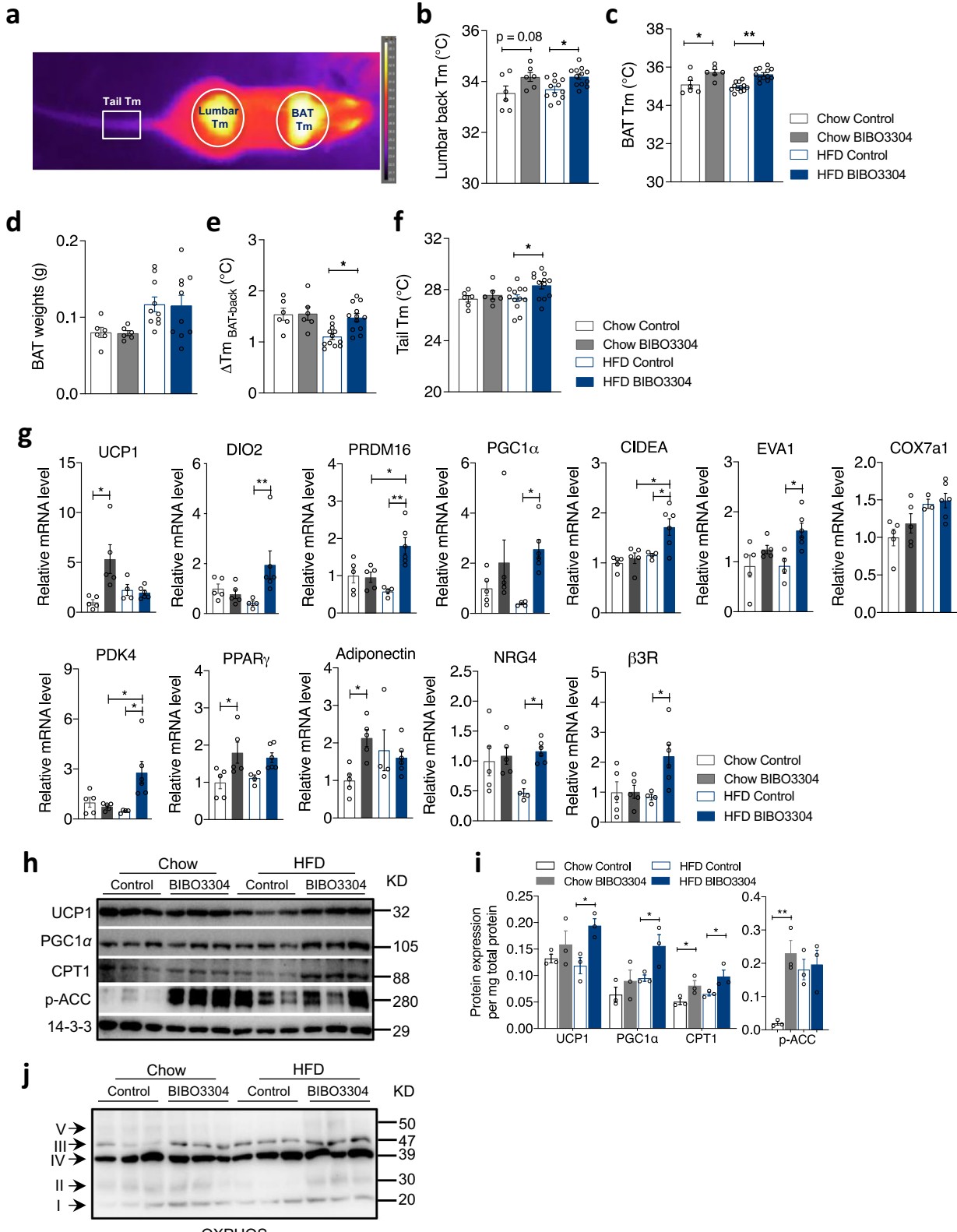

indicating that heat loss through the tail may also contribute to the overall increased EE in response to peripheral selective Y1R antagonism.

**Peripheral selective Y1R signalling controls BAT activity.** In order to gain insights on how Y1R signalling may influence BAT thermogenesis, we measured the expression level of some key BAT transcripts by quantitative PCR (qPCR) as well as western blotting. Under a chow diet, BIBO3304 treatment led to an increase in BAT-specific gene *UCP1* but did not alter the expression of other known thermogenic markers and transcriptional regulators including *DIO2* (type 2 5′ deiodinase), *PRDM16* (PR-domain-containing 16), *PGC1α* (peroxisome proliferator-

**Fig. 4 Peripheral Y1R antagonism increases thermogenesis and promotes the activity of brown adipose tissue. a** Representative infra-red thermal image of the brown adipose tissue (BAT), lumbar back and tail regions of a mouse in this study. Tm Temperature. **b** Temperature measured by a highly sensitive infra-red camera of lumbar back regions of chow- and HFD-fed mice given vehicle or BIBO3304 jelly at 11 weeks of age. **c** Intrascapular BAT temperature measured by infra-red camera from the same set of mice. Data (**b**, **c**) are mean ± s.e.m, chow $n = 6$ (control: open grey; BIBO3304: grey), HFD $n = 12$ (control: open blue; BIBO3304: blue), $*p < 0.05$, $**p < 0.01$, one-way ANOVA with Tukey's multiple comparisons test. **d** Dissected BAT weights of vehicle- and BIBO3304-treated mice on a chow or a HFD for 7 weeks at cull. Data are mean ± s.e.m, chow $n = 6$, HFD $n = 10$, $p$ value by one-way ANOVA with Tukey's multiple comparisons test. **e** The temperature differences between the interscapular and lumbar regions in mice administrated with a jelly containing either vehicle or BIBO3304. **f** Tail temperature measured by a thermal camera in the same cohorts of mice. Data (**e**, **f**) are mean ± s.e.m, chow $n = 6$, HFD $n = 12$, $*p < 0.05$, $**p < 0.01$, one-way ANOVA with Tukey's multiple comparisons test. **g** mRNA levels of thermogenic and lipogenic markers in intrascapular BAT of chow- and HFD-fed wild-type mice on a daily administration with either control or BIBO3304 jelly. Data are mean ± s.e.m, chow $n = 5$ (control: open grey; BIBO3304:grey). HFD control $n = 4$ (open blue), HFD BIBO3304 $n = 6$ (blue). $*p < 0.05$, $**p < 0.01$, two-way ANOVA with Sidak's multiple comparisons test. **h** Western blot images of UCP1, PGC1α, CPT1 and p-ACC protein levels in BAT of vehicle- or BIBO3304-treated mice. p phosphorylated. 14-3-3 was used as a loading control. Images are representative of 3 independent experiments. KD kilodalton. **i** Quantification of the protein levels normalized to total protein content in BAT or WATi. Data are mean ± s.e.m, $n = 3$ per group (chow, control: open grey; BIBO3304: grey. HFD, control: open blue; BIBO3304: blue), $*p < 0.05$, $**p < 0.01$, one-way ANOVA with Tukey's multiple comparisons test. **j** Representative western blotting image of protein levels of 5 OXPHOS complexes in mitochondrial respiration in mice on a chow or a HFD treated with vehicle or BIBO3304. $n = 3$ per group. Source data are provided as a Source Data file.

activated receptor γ coactivator 1α), *CIDEA* (cell death-inducing DNA fragmentation factor alpha-like effector A), *EVA1* (epithelial V-like antigen 1), *Cox7a1* (cytochrome C oxidase subunit 7a1) and *PDK4* (pyruvate dehydrogenase kinase 4) (Fig. 4g). Interestingly, the expression of adipocyte-specific genes such as *PPARγ* and *adiponectin* were also significantly elevated in BAT of chow-fed mice treated with BIBO3304 (Fig. 4g). By contrast, under high fat feeding a distinct pattern of elevated thermogenic markers appeared only in the BIBO3304-treated group. *UCP1* mRNA was not altered by BIBO3304 treatment in HFD-fed mice, but the mRNA levels of a number of other critical thermogenic markers, including *DIO2*, *PRDM16*, *PGC1α*, *CIDEA*, *EVA1* and *PDK4* were markedly upregulated (Fig. 4g). The expression of BAT enriched *NRG4* (neuregulin 4) as well as *β3R* (β3 adrenergic receptor) mRNA were also significantly higher in BIBO3304-treated BAT under a HFD, but were not affected by a chow diet (Fig. 4g). Furthermore, western blot analysis revealed that consistent with a trend in elevated *UCP1* mRNA expression, UCP1 as well as PGC1α protein levels in BAT of BIBO3304-treated mice was upregulated under both dietary conditions (Normalized to total protein content per BAT, Fig. 4h, i), significantly so in the HFD group (Fig. 4i). In keeping with the observed enhanced thermogenesis, CPT1 (carnitine palmitoyltransferase 1), a critical rate-limiting transmembrane enzyme governing the transport of lipids into the mitochondria for oxidation, was significantly elevated with BIBO3304 treatment under both chow and HFD conditions (Fig. 4h, i), thereby likely providing more fuel to the mitochondria. In addition, phosphorylated ACC (acetyl CoA carboxylase) levels were increased in BIBO3304-treated BAT (Fig. 4h, i), suggesting that the biosynthesis of fatty acids is inhibited by BIBO3304 treatment. In line with the observed increased uncoupling of the mitochondrial electron transport chain, the protein levels of several enzyme complexes, notably Complex III, II and I, were higher in BIBO3304-treated BAT than control BAT under both diets (Fig. 4j).

**Peripheral specific Y1R antagonism promotes beige fat development.** Since BAT is relatively smaller in size compared to WAT in rodents, we investigated the possibility of altered function of WAT by peripheral Y1R blockade. Because browning of WAT has been linked to increased thermogenesis, EE and improvement of glucose tolerance[19], and it is more likely to occur in subcutaneous fat depots such as in inguinal adipose tissue (WATi)[20], the largest fat depot in mice and humans, we examined mRNA expression of a number of key signature brown fat-related genes in WATi. Under a chow diet, there was only low mRNA

expression of brown fat-specific and beige fat-selective markers including *UCP1*, *DIO2*, *CIDEA*, *COX7a1*, *PDK4*, *PRDM16*, *EVA1* and *FBXO31*, whose expression was also not affected by BIBO3304 treatment except for *PGC1α* whose expression was upregulated by Y1R blockade (Fig. 5a). Interestingly, the mRNA expression of a subset of beige-selective genes such as *CD137*, *Tmem26*, *Klhl13* and *Slc27a1*, but not *Ebf3*, *Acot2* and *Slc29a1*, was induced by Y1R blockade in WATi under a chow condition (Fig. 5b), indicating the recruitment of beige adipocytes in WATi under this condition. The mRNA expression of *PPARγ* and *adiponectin*, was also upregulated in WATi in chow-fed BIBO3304-treated mice (Fig. 5c). By stark contrast, the expression of thermogenic genes such as *UCP1*, *CIDEA*, *COX7a1*, *Tbx1*, *PGC1α*, *Slc27a1* and *IRF4* was markedly reduced by HFD feeding (Fig. 5a–c). In particular, *UCP1* mRNA expression in WATi was abolished by HFD feeding, but was significantly elevated by BIBO3304 treatment relative to HFD-fed controls (Fig. 5a). Moreover, a marked increase in mRNA expression of brown fat-selective genes and beige signature genes including *DIO2*, *CIDEA*, *COX7a1*, *Tbx1*, *PDK4*, *PGC1α*, *Acot2* and *CD137* was observed in WATi of BIBO3304-treated mice (Fig. 5a). No significant difference was seen with regards to the mRNA expression of *PRDM16*, *EVA1*, *FBXO31*, *Ebf3*, *Tmem26*, *Klhl13*, *Scl27a1* and *Scl29a1* by BIBO3304-treated DIO mice (Fig. 5a, b). Furthermore, mRNA expression of *adiponectin* and *β3R*, but not *PPARγ*, was also increased by BIBO3304 treatment under a HFD (Fig. 5c).

Consistent with the mRNA data, protein levels of UCP1 and PGC1α showed nonsignificant trends towards an increase in WATi of BIBO3304-treated mice (Normalized to total protein content per WATi, Fig. 5d), quantified in Fig. 5e. In addition, p-ACC protein expression tended to elevate under chow feeding but not under a HFD (Fig. 5d, e). Of note, HFD feeding significantly reduced p-ACC expression and also led to a nonsignificant trend towards a decreased protein expression of UCP1 and PGC1α (Fig. 5d, e). Moreover, immunohistochemistry analysis confirmed that UCP1 expression was induced in WATi tissue of BIBO3304-treated mice fed with a HFD (Fig. 5f).

Next, to verify that the observed altered expression of thermogenic markers in BIBO3304-treated mice is a direct result of the lack of Y1R signalling, we harvested WATi from germline Y1R[−/−] mice and performed qPCR on isolated mRNA. Importantly, the mRNA expression of *UCP1*, *DIO2*, *CIDEA* and *Cox7a1* in WATi of Y1R[−/−] mice was significantly higher compared to that in WATi of WT mice (Fig. 5g), confirming the results obtained from the pharmacological approach using BIBO3304. To further confirm that thermogenic effects induced

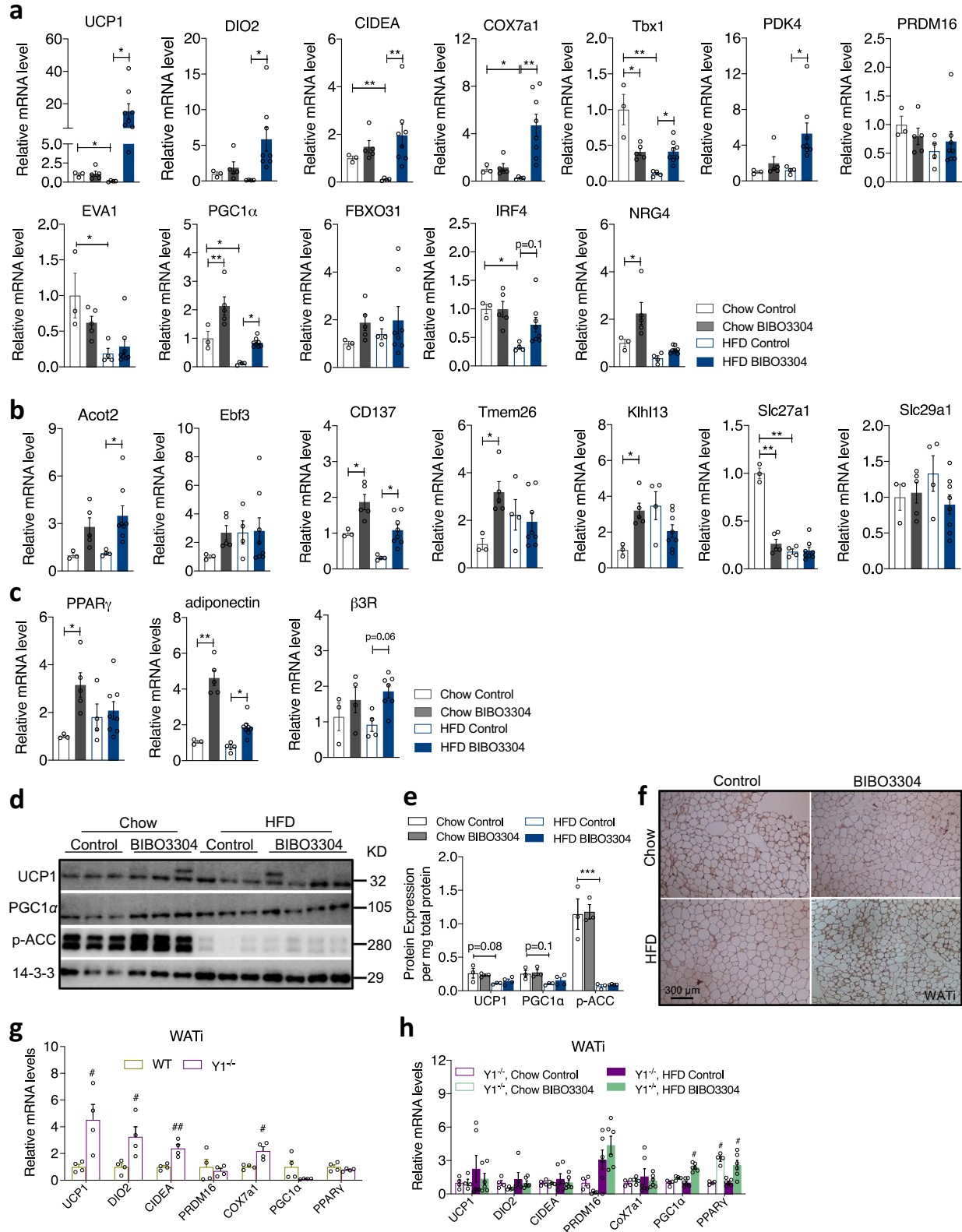

by BIBO3304 is primarily Y1R dependent, chow and HFD-fed Y1R$^{-/-}$ mice were treated with BIBO3304 and qPCR was performed on isolated mRNA from WATi. In the absence of Y1R, the upregulated expression of BAT-selective markers observed in BIBO3304-treated wild type mice (Fig. 5a) did not occur in BIBO3304-treated Y1R$^{-/-}$ mice except for *PGC1α* and *PPARγ*, which may also be regulated by Y1R-independent mechanisms

(Fig. 5h). These data further demonstrate that the lack of Y1R signalling in WATi is critical for regulating the expression of thermogenic genes. Consistent with this, BIBO3304-treated HFD-fed Y1R$^{-/-}$ mice displayed no difference from control-treated Y1R$^{-/-}$ mice with regards to body weight and fat mass (Supplementary Fig. 3a–c). Glucose tolerance was also similar between

**Fig. 5 Peripheral Y1R antagonism induces browning of white adipose tissue. a** The mRNA levels of thermogenic and lipogenic markers in inguinal WAT (WATi) of chow- and HFD-fed wild-type mice on a daily administration with a jelly containing a vehicle or BIBO3304. **b** mRNA levels of beige adipocyte-related genes in WATi. **c** mRNA expression of *PPARγ*, *adiponectin*, and *β3R* in WATi of chow- and HFD-fed wild-type mice on a daily administration with a jelly containing vehicle or BIBO3304. Data (**a**, **b**, **c**) are mean ± s.e.m, $n = 3$ in chow control (open grey), and 5 in chow BIBO3304 (grey); $n = 4$ in HFD control (open blue), and 8 in HFD BIBO3304 (blue). *$p < 0.05$, **$p < 0.01$, two-way ANOVA with Sidak's multiple comparisons test. **d** Western blot analyses of UCP1, PGC1α and p-ACC protein levels in the WATi of vehicle- or BIBO3304-treated mice. 14-3-3 was used as a loading control. Images are representative of three independent experiments. KD kilodalton. **e** Quantification of the protein levels normalized to total protein content in BAT or WATi. Data are mean ± s.e.m, $n = 3$ in chow (control: open grey; BIBO3304: grey) or 4 in HFD (control: open blue; BIBO3304: blue). ***$p < 0.001$, two-way ANOVA with Sidak's multiple comparisons test. **f** Representative immunohistochemical images of UCP1 protein expression in WATi of wild-type mice on either vehicle or BIBO3304 jellies for 7 weeks. $n = 3$ per group. **g** The mRNA levels of thermogenic and adipogenic markers in inguinal WAT (WATi) of Y1R$^{-/-}$ mice compared to wild type mice. Data are mean ± s.e.m, $n = 4$ per group (WT: mustard; Y1R$^{-/-}$: purple). #$p < 0.05$, ##$p < 0.01$, Y1R$^{-/-}$ versus wild type determined by two-tailed *t* test. **h** The mRNA levels of thermogenic genes in WATi of chow- and HFD-fed Y1R$^{-/-}$ mice given either vehicle jelly or BIBO3304 jelly daily for 7 weeks. Data are mean ± s.e.m, chow $n = 4$ (control: open purple; BIBO3304: open green), HFD $n = 6$ (control: purple; BIBO3304: green). #$p < 0.05$, ##$p < 0.01$, two-way ANOVA with Sidak's multiple comparisons test. Source data are provided as a Source Data file.

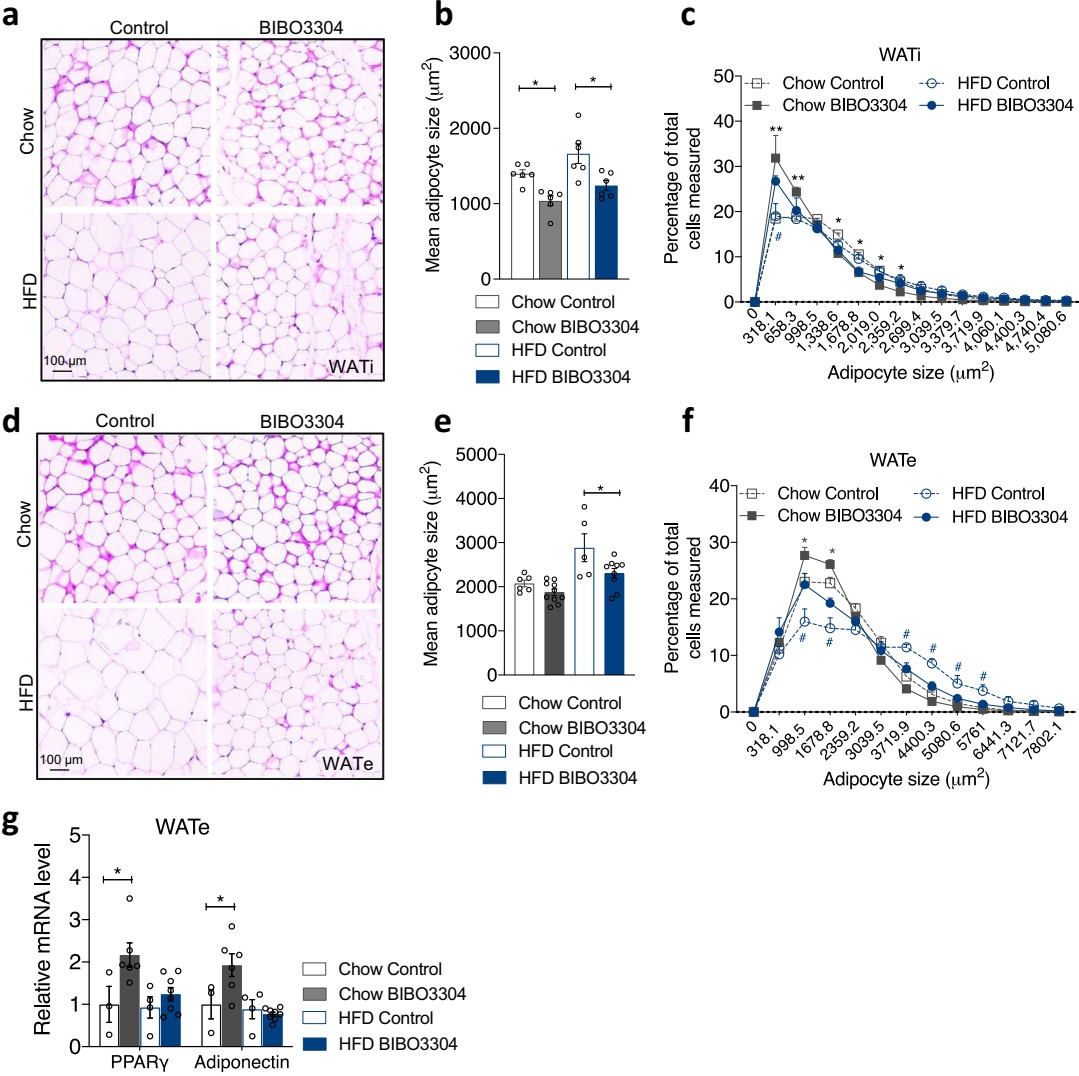

**Fig. 6 Effects of peripheral Y1R antagonism on adipocyte size and distribution. a d** H&E staining of WATi and WATe of vehicle- and BIBO3304-treated wild-type mice on either a chow or a HFD. Images are representative of three independent experiments. **b**, **e** Average adipocyte size of WATi and WATe in BIBO3304-treated mice relative to that in vehicle-treated mice. Data are mean ± s.e.m, $n = 4$ per group (chow, control: open grey; BIBO3304: grey. HFD control: open blue; BIBO3304: blue). *$p < 0.05$, one-way ANOVA with Tukey's multiple comparisons test. **c**, **f** The frequency distribution of adipocyte size in WATi and WATe from vehicle- and BIBO3304-treated mice on a chow or a HFD. Data are mean ± s.e.m, $n = 6$ per group (chow, grey; HFD, blue. control: open square; BIBO3304: filled square). *$p < 0.05$, **$p < 0.01$ BIBO3304 vs control in chow; #$p < 0.05$ BIBO3304 vs control in HFD, two-way repeated measures ANOVA. **g** The mRNA levels of adipogenic markers in epididymal WAT (WATe) of chow- or HFD-fed wild-type mice treated with vehicle or BIBO3304 jellies. Data are mean ± s.e.m, chow: control $n = 3$ (open grey), BIBO3304 $n = 6$ (grey); HFD: control $n = 4$ (open blue), BIBO3304 $n = 8$ (blue). *$p < 0.05$, two-way ANOVA with Sidak's multiple comparisons test. Source data are provided as a Source Data file.

BIBO3304 and vehicle-treated Y1R$^{-/-}$ mice (Supplementary Fig. 3d). Taken together, these results further confirm that Y1R signalling is required for the thermogenic actions of BIBO3304 treatment to achieve metabolic benefits on body weight and fat mass.

**Reduced adipocyte size due to peripheral selective Y1R antagonism.** To assess whether the increased thermogenesis in BAT and WAT browning also led to a change in adipocyte phenotype, we performed morphological examination on WATi and WATe tissue. hematoxylin and eosin (H&E) staining showed WATi adipocytes of BIBO3304-treated mice fed with a HFD were generally smaller than those in HFD-fed control animals (Fig. 6a) which is confirmed by quantification (Fig. 6b). Further analysis of adipocyte distribution also revealed that there was a significant increase in the proportion of smaller adipocytes in the BIBO3304-treated mice (Fig. 6c). Similarly, the average size of adipocytes in WATe of HFD-fed BIBO3304-treated mice were smaller than those in vehicle-treated WATe (Fig. 6d, e). A significant increase in the percentage of smaller adipocytes in WATe from BIBO3304-treated mice was also observed (Fig. 6f). Furthermore, this reduced average adipocyte size and increased proportion of smaller cells in WATe of BIBO3304-treated mice was associated with upregulated *PPARγ* and *adiponectin* mRNA levels (Fig. 6g). However, the mRNA expression of a subset of the BAT-specific and beige-selective genes in WATe was unaffected by BIBO3304 treatment except for *CD137* and *NRG4* under a chow diet (Supplementary Fig. 4a, b). These findings suggest that while some beneficial effects can be observed, WATe is less responsive to browning stimuli induced by Y1R antagonism compared to WATi.

**UCP1 is required for metabolic benefits induced by peripheral selective Y1R antagonism.** Since BIBO3304-induced metabolic benefits were associated with marked upregulation of UCP1 expression in BAT and WATi of WT mice (Figs. 4g, 5a), we next assessed whether these effects were specifically dependent on the presence of UCP1. For this, BIBO3304 or vehicle was administrated to UCP1$^{-/-}$ mice under HFD conditions following the same protocol as above. Importantly, we found that the inhibitory effect on weight gain induced by BIBO3304 treatment in HFD-fed WT mice was no longer observed in UCP1$^{-/-}$ mice (Fig. 7a, b). Similarly, fat mass determined by DEXA (Fig. 7c), dissected WAT weights (Fig. 7d) and lean mass (Fig. 7e) did not differ between the two treatment groups. These data suggest that the beneficial effects of BIBO3304 on body weight and fat mass reduction are largely dependent on UCP1. EE normalized to MATs was similar between two treatment groups in UCP1$^{-/-}$ mice (Fig. 7f, g). Likewise, RER was not different between the two treatment groups (Fig. 7h, i), suggesting again UCP1 is required for Y1R antagonism-induced EE and RER. Moreover, analysis of BAT temperature by thermal camera revealed that BIBO3304-treated UCP1$^{-/-}$ mice displayed similar BAT and lumbar back temperature as vehicle-treated mice (Fig. 7j). Furthermore, glucose tolerance was not significantly different between BIBO3304- and vehicle-treated UCP1$^{-/-}$ mice (Fig. 7k). Together, these data confirm that the major beneficial effects of peripheral selective Y1R antagonism require a functional response via UCP1 action since these effects were lost in the absence of UCP1.

**Y1R signalling in adipose tissues is critical for obesity development.** Our initial expression profiling revealed that peripheral Y1R expression is most prominent in adipose tissue and that the major changes in Y1R expression induced by HFD occur at this site (Fig. 1a). Therefore, we next investigated whether selective ablation of Y1R in adipocytes in the adult would mimic the pharmacological response to Y1R antagonism. For this, we generated an adult-onset inducible adipocyte-specific Y1R deletion mouse models employing mice with Cre-recombinase driven by an inducible adiponectin promoter (Jackson Laboratory) (Adipo$^{TMCre/+}$;Y1$^{lox/lox}$). Ablation of the Y1R was initiated in 6-week-old mice by three times gavage of tamoxifen (5 μg in 100 μl sunflower oil) at 2-day intervals, with HFD commencing 4 days after the last dose of tamoxifen and continuing for the next 8 weeks (Fig. 8a). To confirm the adipocyte-specific *Y1R* deletion, total RNA was extracted from WATi, WATe and BAT and subjected to qPCR analysis. Results from this analysis clearly demonstrate that the expression of *Y1R* in WATi and WATe from tamoxifen-induced Adipo$^{TMCre/+}$;Y1$^{lox/lox}$ mice was significantly reduced compared to tissue isolated from vehicle-injected control Adipo$^{TMCre/+}$;Y1$^{lox/lox}$ mice (Fig. 8b). *Y1R* mRNA expression in BAT from tamoxifen-induced Adipo$^{TMCre/+}$;Y1$^{lox/lox}$ mice was also markedly decreased (Fig. 8b) whereas *Y1R* mRNA expression in non-adipose tissues, the liver and skeletal muscle, was similar between tamoxifen- and vehicle-treated control mice (Fig. 8b).

To control for the effects that might be caused by tamoxifen, we used 'WT' littermates without Cre (Adipo$^{+/+}$;Y1$^{lox/lox}$) treated with tamoxifen or vehicle. Not unexpected[21], tamoxifen administration in Adipo$^{+/+}$;Y1$^{lox/lox}$ mice led to a marked and significant increase in body weight gain compared to vehicle-treated Adipo$^{+/+}$;Y1$^{lox/lox}$ mice under both chow (Fig. 8c) and HFD conditions (Fig. 8g) when monitored over the following 8-week period. The increase in body weight was mainly due to an increase in fat accumulation (Fig. 8d, h). Importantly, this increase in body weight gain in tamoxifen-treated Adipo$^{+/+}$;Y1$^{lox/lox}$ control mice did not occur in age-matched tamoxifen-treated Y1R deletion Adipo$^{TMCre/+}$;Y1$^{lox/lox}$ mice, which displayed a comparable body weight gain to vehicle-treated Adipo$^{+/+}$;Y1$^{lox/lox}$ or Adipo$^{TMCre/+}$;Y1$^{lox/lox}$ mice under either chow (Fig. 8e) or HFD conditions (Fig. 8i). Furthermore, increased fat mass seen in tamoxifen-treated WT mice was no longer observed in tamoxifen-treated Adipo$^{TMCre/+}$;Y1$^{lox/lox}$ mice on a HFD (Fig. 8j), and was actually significantly reduced in tamoxifen-treated Adipo$^{TMCre/+}$;Y1$^{lox/lox}$ mice compared to vehicle-treated mice on a chow diet (Fig. 8f). These data show that tamoxifen per se can have an anabolic and obesogenic effect, importantly however, the absence of Y1R signalling inhibits this effect by reducing fat mass accumulation, thus further confirming the crucial role of Y1R in the control of body weight and fat accumulation.

Moreover, HFD-fed tamoxifen-treated Adipo$^{+/+}$;Y1$^{lox/lox}$ mice displayed glucose intolerance compared to vehicle-treated Adipo$^{+/+}$;Y1$^{lox/lox}$ mice (Fig. 8k), while tamoxifen-treated Adipo$^{TMCre/+}$;Y1$^{lox/lox}$ mice displayed a comparable glucose excursion to vehicle-treated Adipo$^{TMCre/+}$;Y1$^{lox/lox}$ mice (Fig. 8l). Furthermore, in line with impaired GTT, these tamoxifen-treated Adipo$^{+/+}$;Y1$^{lox/lox}$ control mice showed a tendency to have worsened insulin responsiveness (Fig. 8m). However, this was not seen in tamoxifen-treated Adipo$^{TMCre/+}$;Y1$^{lox/lox}$ mice compared to control mice on the same HFD (Fig. 8n). Collectively, these data support that Y1R signalling in adipose tissues plays a critical role in the regulation of whole-body energy homeostasis; and that antagonism of Y1R in adipocytes is able to improve glucose and energy homeostasis.

**Y1R antagonism activates thermogenesis in primary human adipocytes.** The NPY system is known for being conserved during evolution. However, to test whether the effects of Y1R signalling in adipose tissue is conserved in humans, we collected SAT and intra-abdominal VAT from normal body weight and obese adult subjects, and profiled the mRNA expression of

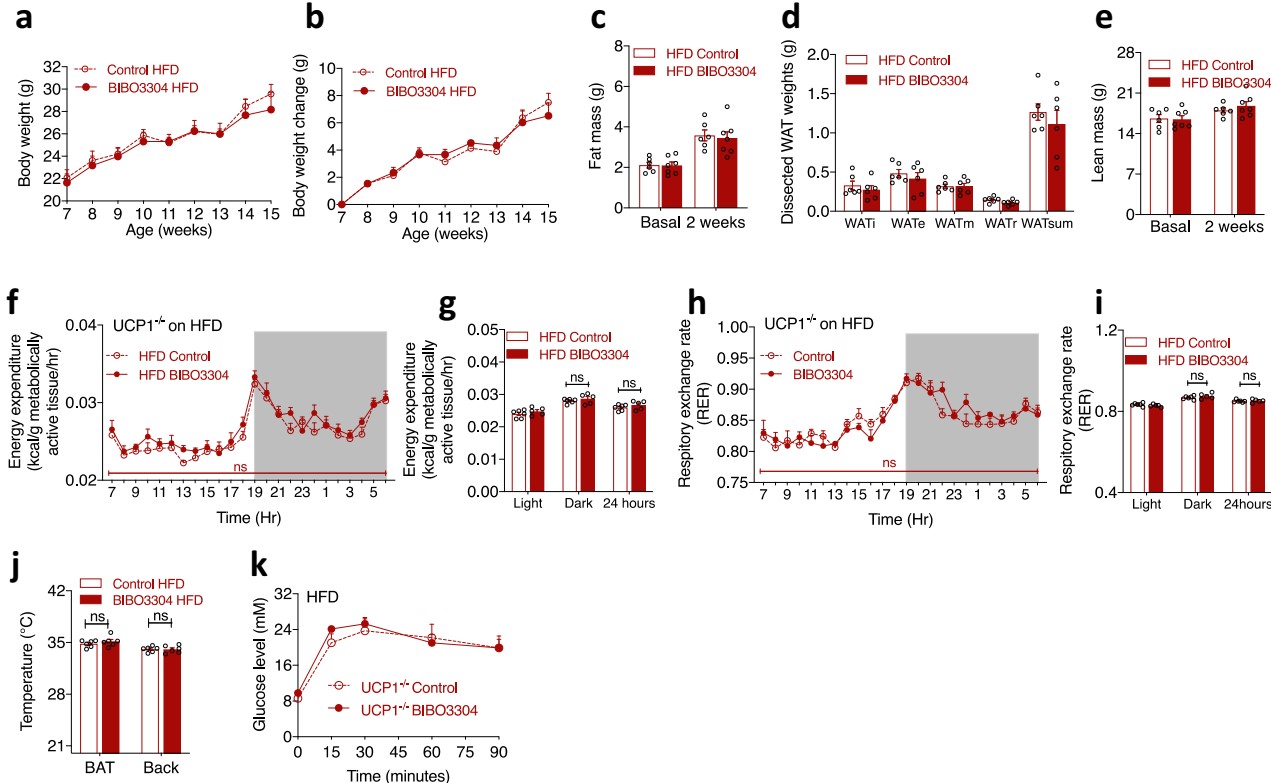

**Fig. 7 UCP1 is required for the weight-loss effects of BIBO3304 on a high-fat diet. a, b** Body weight curve and body weight change of UCP1$^{-/-}$ mice on a high-fat diet administered daily with either a jelly containing vehicle (control) or BIBO3304 for 8 weeks. Data are mean ± s.e.m, $n = 6$ per group (control: open circle; BIBO3304: filled circle). $p$ values by two-way repeated measures ANOVA. **c** Whole-body fat mass of vehicle- and BIBO3304-treated UCP1$^{-/-}$ mice on a HFD determined by DEXA at baseline and after 2 weeks of treatment. **d** Dissected weights of individual WAT from inguinal (WATi), epididymal (WATe), mesenteric (WATm), retroperitoneal (WATr) and total weights of these 4 depots (WATsum) at cull. **e** Whole-body lean mass of vehicle- and BIBO3304-treated UCP1$^{-/-}$ mice on a HFD determined by DEXA at baseline and after 2 weeks of treatment. Data are mean ± s.e.m, $n = 6$ per group (control: open red; BIBO3304: red). $p$ values by two-sided $t$ test (**c**, **d**, **e**). **f**, **g** Energy expenditure, **h**, **i** RER of vehicle- and BIBO3304-treated UCP1$^{-/-}$ mice on a HFD over a 24 h course during an indirect calorimetry study at 13 weeks of age. Data are mean ± s.e.m, control $n = 6$ (open circle), BIBO3304 $n = 5$ (filled circle). $p$ values by two-way repeated measures ANOVA. **j** Temperature measured by infra-red camera of BAT and lumbar back regions of HFD-fed UCP1$^{-/-}$ mice given vehicle or BIBO3304 jelly at 11 weeks of age. Data are mean ± s.e.m, control $n = 6$ (open red), BIBO3304 $n = 5$ (red). $p$ values by two-sided $t$ test within the same region. **k** Glucose levels in an intraperitoneal glucose tolerance test (GTT) in vehicle- and BIBO3304-treated UCP1$^{-/-}$ mice on a HFD at 13 weeks of age. Data are mean ± s.e.m. $n = 6$ per group (control: open circle; BIBO3304: filled circle). $p$ values by two-way repeated measures ANOVA. ns non-significance. Source data are provided as a Source Data file.

thermogenic markers in SAT and VAT by qPCR. Importantly, the mRNA expression of key thermogenic genes, such as *UCP1*, *PGC1α*, *Cidea*, *CD137* and *Tmem26*, as well as adipogenic genes *Adiponectin* and *PPARγ* was markedly downregulated in both SAT (Fig. 9a) and VAT (Fig. 9b) from obese subjects compared to normal weight individuals. The inverse relationship of these genes with the upregulated *Y1R* mRNA expression in adipose tissue in these obese subjects (Fig. 1b) supports the proposition that elevated Y1R signalling may suppress thermogenic activity in human WATs, thereby contributing to the development of obesity.

To test this hypothesis, we isolated primary stromal vascular fractions (SVF), which contain pre-adipocytes from the SAT of obese subjects and differentiated them into mature adipocytes ex vivo and treated them with BIBO3304 in the presence or absence of $^{[Leu31,Pro34]}$NPY, a Y1R preferring agonist for 48 h. mRNA isolated from these cells was then analyzed by qPCR for determining changes in expression of thermogenic gene markers. *NPY* mRNA is expressed in SVF pre-adipocytes and mature differentiated adipocytes (Supplementary Fig. 5). Importantly, we found that compared to saline-treated controls, BIBO3304 treatment in primary human adipocytes led to a significant

upregulation in mRNA expression of *UCP1*, *CD137* and *Tmem26*, but not *PGC1α* and *Cidea* (Fig. 9c), suggesting activation of a transcriptional programme consistent with increased thermogenesis. By contrast, Y1R selective agonist $^{[Leu31,Pro34]}$NPY treatment suppressed *PGC1α* expression when compared with controls (Fig. 9c), which importantly was restored after co-treatment with BIBO3304, except for *UCP1* which may be due to treatment timing issues (Fig. 9c). Collectively, these data from human adipocytes confirm the findings in mice and strongly suggests that Y1R antagonism contributes to the activation of WAT browning also in humans.

**Peripheral specific Y1R blockade improves metabolic outcomes via multiple signalling pathways.** Classically the Y1R interacts with $G_{i/o}$ G-proteins, which act to inhibit adenylate cyclase activity to reduce cAMP accumulation in target cells[22]. To unravel the intracellular components that are important in mediating the action of Y1R signalling on BAT activity and WAT browning, we first performed western blot analysis for p-CREB, a key downstream mediator of the cAMP signalling pathway. BIBO3304 treatment resulted in a significant increase in p-CREB in BAT of

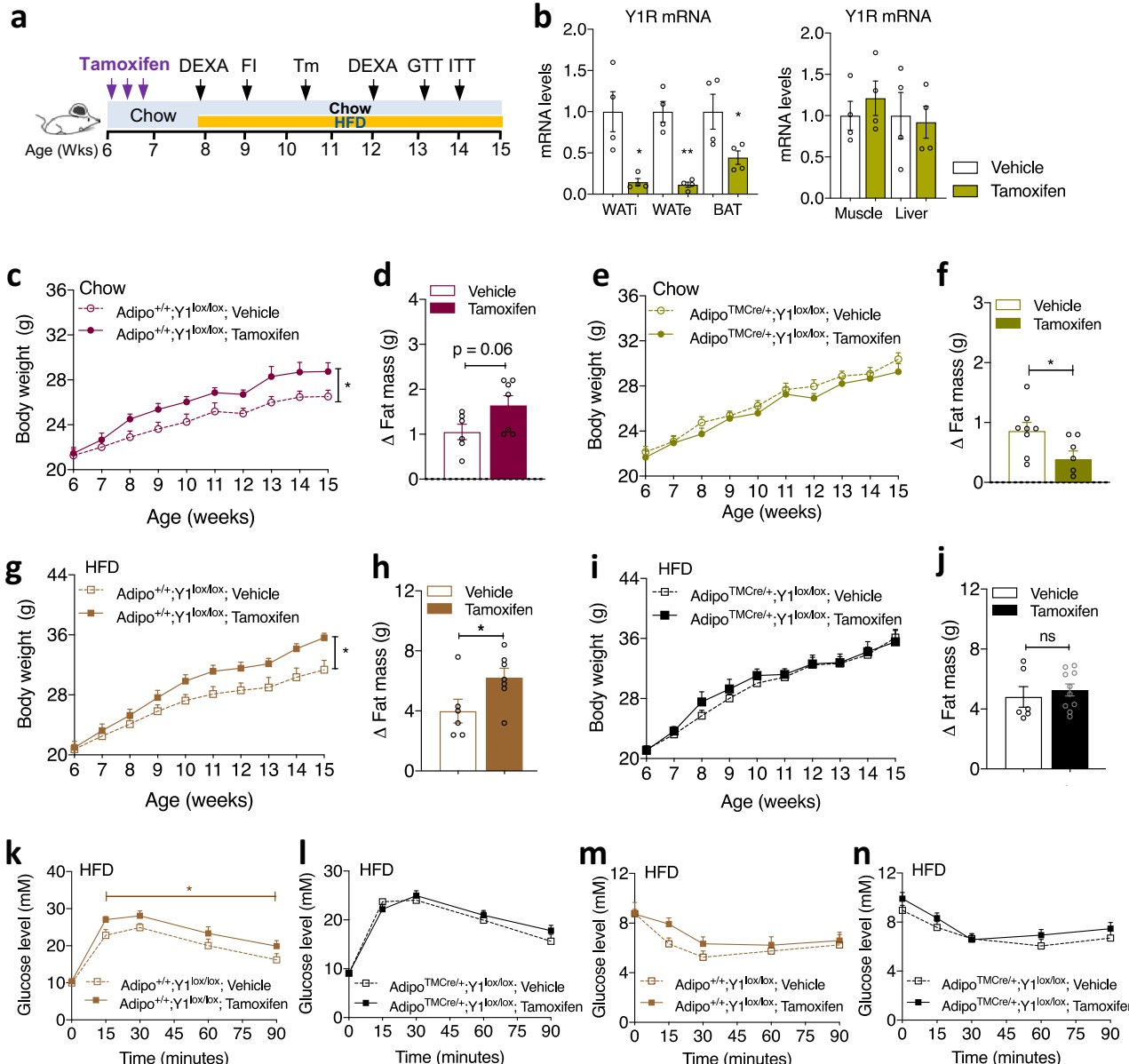

**Fig. 8 Y1R signalling in adipose tissue is critical in obesity development. a** Schematic illustration of the paradigm for phenotyping the tamoxifen-induced adipocyte-specific Y1R deletion mouse model. **b** Relative *Y1R* mRNA expression in inguinal WATi, epididymal WATe, BAT, skeletal muscle and the liver of Adipo$^{TMCre/+}$;Y1$^{lox/lox}$ mice receiving either vehicle or tamoxifen determined by RT-PCR using *RPL19* as a housekeeping gene. Data are mean ± s.e.m. $n = 4$ per group (vehicle: white; tamoxifen: mustard). *$p < 0.05$ vs vehicle in the same fat depot by two-tailed $t$ test. **c, g** Body weight of vehicle- or tamoxifen-treated Adipo$^{+/+}$;Y1$^{lox/lox}$ control mice on a standard chow diet or a HFD for 8 weeks. Data are mean ± s.e.m. Chow (**c**): vehicle $n = 7$ (open circle), tamoxifen $n = 6$ (filled circle); HFD (**g**): vehicle $n = 6$ (open square), tamoxifen $n = 7$ (filled square). *$p < 0.05$, two-way repeated measures ANOVA. **d, h**, Change of fat mass of vehicle- or tamoxifen-treated Adipo$^{+/+}$;Y1$^{lox/lox}$ mice on a chow diet or HFD determined by DEXA at 12 weeks of age. Data are mean ± s.e.m. vehicle $n = 6$ (open bar), tamoxifen $n = 7$ (filled bar) in both chow (**d**) and HFD (**h**); *$p < 0.05$ by two-tailed t test. **e, i** Body weight of vehicle- or tamoxifen-treated Adipo$^{TMCre/+}$;Y1$^{lox/lox}$ on a chow diet or HFD for 8 weeks. Data are mean ± s.e.m. Chow (**e**): vehicle $n = 8$ (open circle), tamoxifen $n = 10$ (filled circle); HFD (**i**): $n = 7$ per group (vehicle: open square; tamoxifen: filled square), $p$ values by two-way repeated measures ANOVA. **f, j** Change of fat mass of vehicle- or tamoxifen-treated Adipo$^{TMCre/+}$;Y1$^{lox/lox}$ mice on a chow or HFD determined by DEXA at 12 weeks of age. Data are mean ± s.e.m. Chow (**f**): vehicle $n = 7$ (open bar), tamoxifen $n = 6$ (filled bar); HFD (**j**): vehicle $n = 6$ (open bar), tamoxifen $n = 10$ (filled bar). *$p < 0.05$, two-tailed t test. **k, l** Glucose levels in an intraperitoneal glucose tolerance test (GTT) in Adipo$^{+/+}$;Y1$^{lox/lox}$ mice (light brown, vehicle $n = 7$, open square; tamoxifen $n = 8$ filled square) or Adipo$^{TMCre/+}$;Y1$^{lox/lox}$ mice (black, vehicle $n = 7$, open square; tamoxifen $n = 12$ filled square) on a HFD at 13 weeks of age. Data are mean ± s.e.m. *$p < 0.05$, two-way ANOVA. **m, n** Glucose levels in an insulin tolerance test (ITT) in Adipo$^{+/+}$;Y1$^{lox/lox}$ (light brown, vehicle $n = 7$, tamoxifen $n = 8$) or Adipo$^{TMCre/+}$;Y1$^{lox/lox}$ mice (black, vehicle $n = 7$, tamoxifen $n = 11$) on a HFD at 14 weeks of age. Data are mean ± s.e.m. $p$ values by two-way ANOVA. ns non-significance. Source data are provided as a Source Data file.

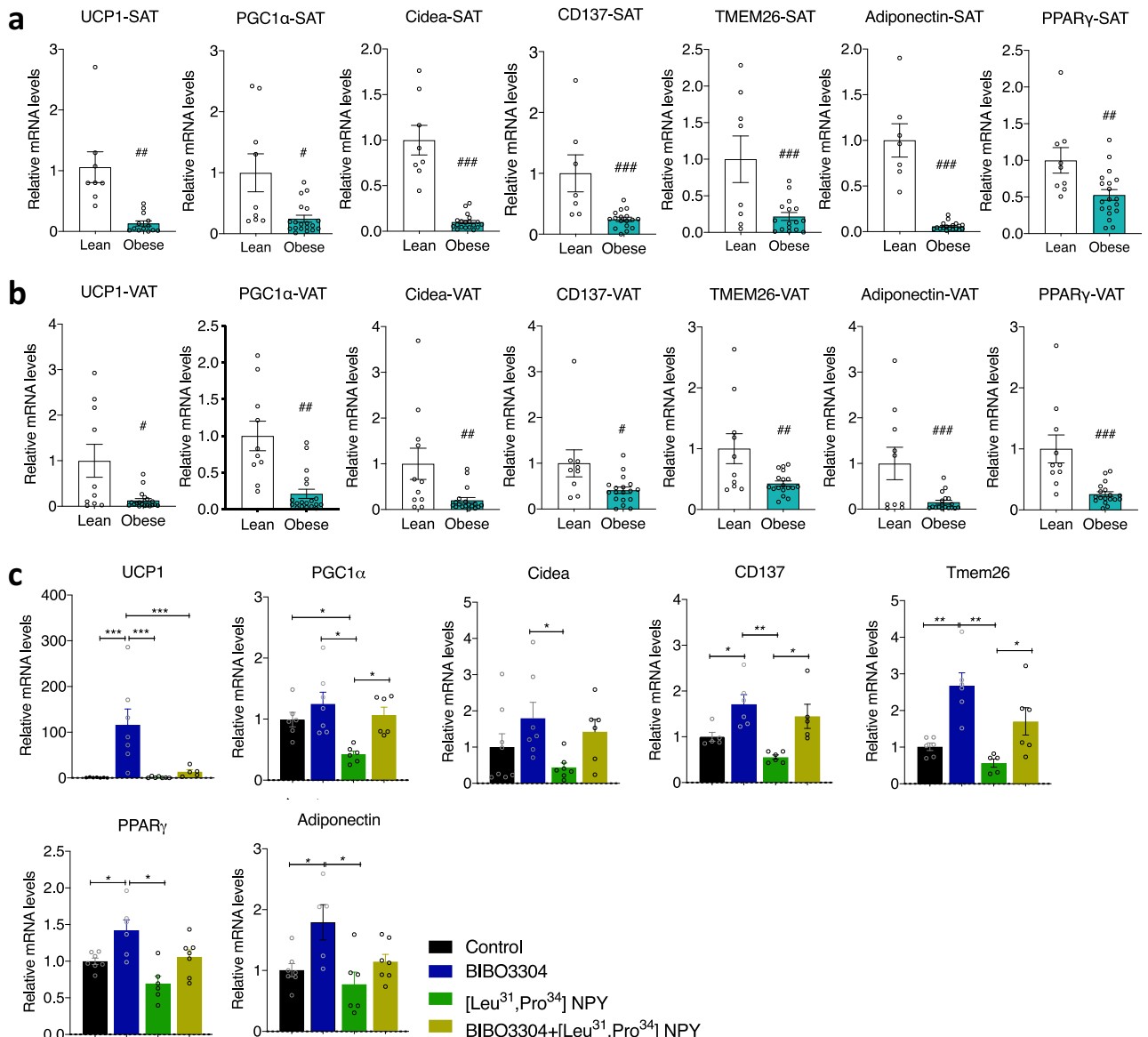

**Fig. 9 Peripheral Y1R antagonism activates thermogenesis in human adipocytes. a** mRNA levels of thermogenic and lipogenic markers in subcutaneous white adipose tissue (SAT) of lean and obese subjects. Data are mean ± s.e.m, lean $n = 9$ (white bar), obese $n = 18$ (cyan bar), #$p < 0.05$, ##$p < 0.01$, ###$p < 0.001$ by two-tailed $t$ test. **b**, mRNA levels of thermogenic and lipogenic markers in visceral adipose tissue (VAT) of lean and obese subjects. Data are mean ± s.e.m, lean $n = 10$ (white bar), obese $n = 19$ (cyan bar). #$p < 0.05$, ##$p < 0.01$, ###$p < 0.001$ by two-tailed t test· **c** mRNA levels of thermogenic and lipogenic markers in stromal vascular fraction (SVF) isolated from obese subjects cultured and differentiated ex vivo and treated with Y1R-specific agonist [Leu31,Pro34]NPY and/or antagonist BIBO3304 for 48 h. Data are mean ± s.e.m, $n = 5$–8 per group (control: black; BIBO3304: blue; [Leu31,Pro34]NPY: green; BIBO3304 + [Leu31,Pro34]NPY: mustard). *$p < 0.05$, **$p < 0.01$, ***$p < 0.001$, one-way ANOVA with Tukey's multiple comparisons test. Source data are provided as a Source Data file.

both chow- and HFD-fed mice (Fig. 10a, b). Similarly, treatment of mice under a chow condition with BIBO3304 upregulated p-CREB in WATi, however, the increase in p-CREB was abolished in WATi of HFD-fed mice (Fig. 10c, d). These data indicate that Y1R signalling does influence this pathway in both BAT and WAT, but demonstrate a tissue-specific regulatory pattern with a HFD. The MAPK-ERK signalling pathways have also been implicated to play a role in the activation of BAT and WAT browning[23]. p-ERK protein levels were also elevated in BAT upon BIBO3304 treatment under a chow diet but not a HFD (Fig. 10a, b), whereas BIBO3304 treatment increased p-ERK in WATi under both chow and HFD conditions (Fig. 10c, d). Together these findings demonstrate cAMP-CREB and MAPK-ERK signalling pathways are primarily

being influenced by Y1R signalling and their actions are enhanced by Y1R signalling blockade.

To further investigate cellular signalling pathways that are activated by Y1R signalling and how they change upon BIBO3304 treatment in vivo, we employed an Akt-FRET biosensor mouse model (validated previously[24,25]) and monitored the dynamic changes of Akt phosphorylation, a central node of metabolic signalling, in BAT by multiphoton microscopy in live animals. This was achieved via implantation of an optical window over the BAT areas in mice (Fig. 10e), which were fed a HFD in the presence or absence of BIBO3304 for 4 weeks. Mice were imaged by fluorescence lifetime imaging of the donor fluorophore mTurquoise2. When Akt is active within the fat cells, the

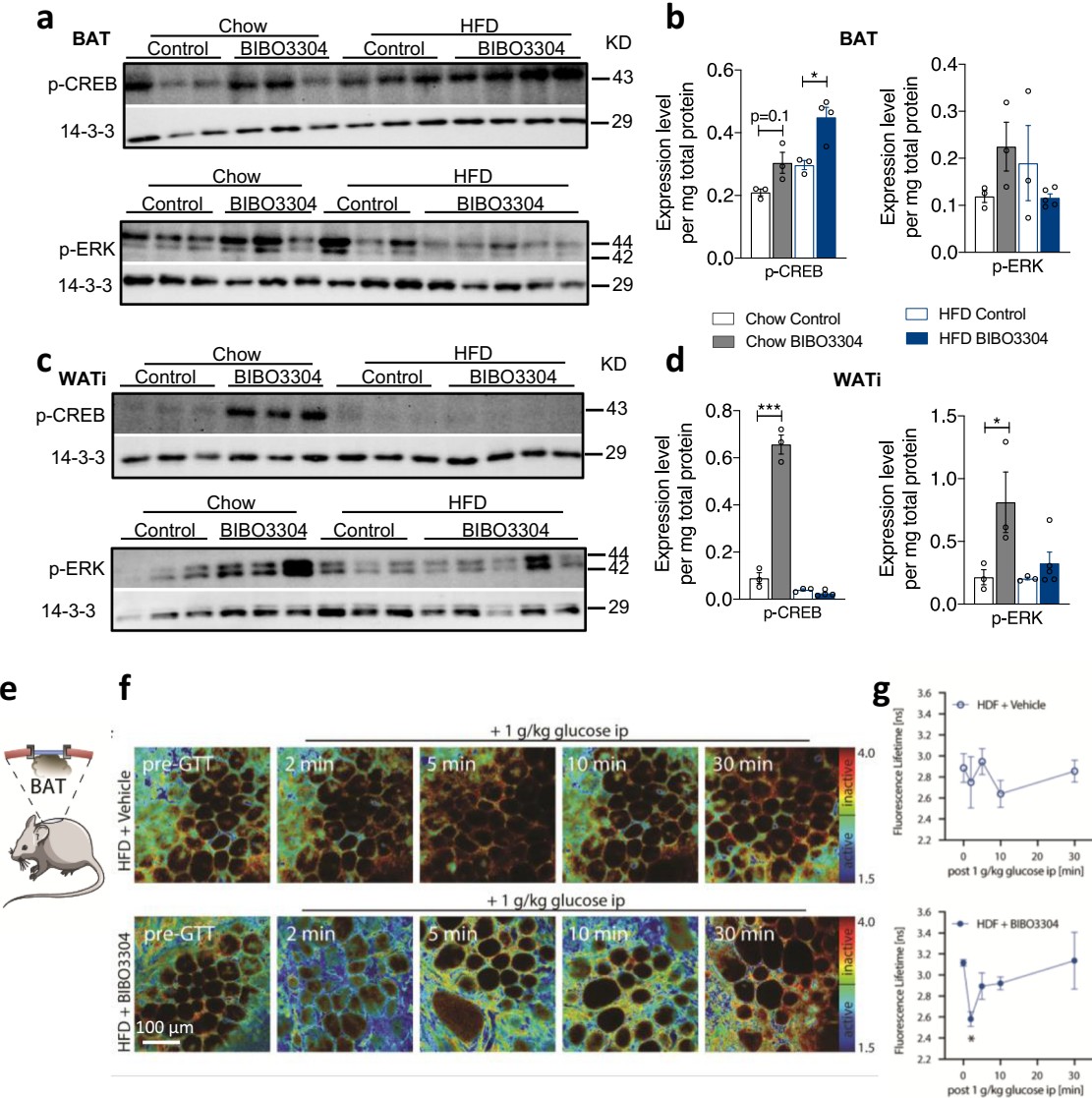

**Fig. 10 Peripheral specific Y1R blockade improves metabolic outcomes via multiple signalling pathways. a, c** Representative western blotting images of p-CREB and p-ERK protein levels in intrascapular BAT and inguinal WAT (WATi) of vehicle- or BIBO3304-treated wild-type mice for 8 weeks. p phosphorylated. 14-3-3 was used as a loading control. Images are representative of three independent experiments. **b, d** Phosphorylated-CREB (p-CREB) and p-ERK protein expression expressed as fold change to housekeeping gene 14-3-3 and relative to chow-fed control in vehicle- or BIBO3304-treated wild type mice for 7 weeks. Data are mean ± s.e.m, chow $n = 3$ (control: open grey; BIBO3304: grey); HFD control $n = 3$ (open blue), BIBO3304 $n = 4$ (p-CREB) or 5 (p-ERK) (blue). *$p < 0.05$, ***$p < 0.001$, two-way ANOVA with Sidak's multiple comparisons test. **e** Schematic of optical window implanted over intrascapular BAT of Akt-FRET biosensor mice. Illustration was adapted from Servier Medical Art, licensed under the Creative Commons Attribution 3.0 Unported license. **f** Representative images of dynamic Akt activity changes after acute glucose challenge in BAT of Akt-FRET biosensor mice on a HFD treated with vehicle or BIBO3304 jelly for 4 weeks, and fluorescence lifetime is quantified in (**g**) unit: nanosecond (ns). Data are mean ± s.e.m, vehicle $n = 4$, BIBO3304 $n = 3$. *$p < 0.05$ compared to 0 time point, Kruskal–Wallis one-way ANOVA. Images are representative of three independent experiments. Source data are provided as a Source Data file.

biosensor assumes a closed conformation resulting in FRET by the juxtaposition of the mTurquoise2 to its acceptor fluorophore YPet. When measuring the fluorescence lifetime in nanoseconds by time-correlated single-photon counting, as previously achieved[25], this will result in a shorter lifetime of mTurquoise2 fluorophore. If Akt is inactivated in cells this lifetime will increase. When we challenged these mice with an i.p. glucose bolus, and visualized real-time Akt activity over the time course of 30 min, we found that compared to control BAT, significantly more Akt was active in BIBO3304-treated BAT (Fig. 10f, g). Active Akt is mapped in the fluorescence lifetime colour heatmaps by blue to green pixels and inactive Akt by yellow to

red. High Akt activity was observed, as evidenced by a pronounced decrease in fluorescence lifetime reflected by the amount of blue signal, which peaked at 2 min after glucose injection and maintained a trend towards a higher level of activity than control BAT throughout the 30-min monitoring period (Fig. 10f, g). More importantly, after BAT isolated from chow-fed mice was treated ex vivo with a mTOR1/2 inhibitor, AZD2014[25] for 30 min, the Akt activity was effectively inhibited (Supplementary Fig. 6), further confirming the specificity of the FRET probe. These data suggest that BIBO3304 treatment enhances insulin signalling in BAT under HFD conditions.

## Discussion

This study demonstrates that development of obesity is promoted by signalling through the Y1R in the periphery, since selective antagonism of peripheral Y1R or deletion of adipose tissue Y1R prevents the development of high-fat DIO. This protective effect of peripheral Y1R signalling antagonism is mainly due to enhanced EE and increased whole-body thermogenesis leading to a reduction in fat mass. Especially enhanced thermogenesis in BAT and extensive browning of WAT contributes to this metabolic benefit. Mechanistically this is due to the significant upregulation of UCP1 as well as a panel of other thermogenic markers in both BAT and inguinal WAT, most likely mediated by the cAMP-CREB and MAPK-ERK pathways. Consistent with this, lack of UCP1 results in a loss of the beneficial effects of Y1R antagonism, as does the selective ablation of Y1R from adipocytes. Importantly, peripheral Y1R antagonism also improves glucose homeostasis by disinhibiting pathways that promote p-Akt activity in adipose tissues. Importantly, these functions of Y1R signalling in adipocytes are evolutionarily conserved and also fully functional in human adipose tissue.

NPY is a stress molecule released into circulation from the adrenal medulla and sympathetic nerves where it serves an essential protective role during acute stress[26]. However, the actions of NPY under prolonged chronic stress conditions can be detrimental[27,28]. Elevated NPY level in the circulation and various adipose tissue sites has been linked to the development of obesity and metabolic diseases in mice[29] and humans[13-15,30]. In fact, NPY is locally produced by pre-adipocytes and mature differentiated adipocytes in humans and mice, with increased expression of NPY and Y1R in adipose tissue favouring the storage of lipids over thermogenesis[9,31]. This NPY-induced storage of energy is an important survival mechanism in mice and humans during times of energy shortage; however, with current overnutrition and sedentary behaviour, this powerful energy conservation capacity can translate into comorbidities, exacerbating already existing diet-induced weight gain and fat accretion. In contrast, the lack of NPY in mice is shown to protect from DIO[32].

The current study shows that periphery specific antagonism of Y1R signalling via the non-brain penetrable Y1R antagonist BIBO3304 prevents DIO via stimulating EE and whole-body thermogenesis. The marked rise in EE can be explained by an increase in thermogenesis in BAT as well as substantial WAT browning, leading to the increased body and BAT temperature. Y1R antagonism powerfully promotes the conversion of energy-storing WATi to energy-burning beige WATi, particularly under energy excess conditions. This is evidenced by ~20-fold increase in UCP1 expression as well as the upregulation of a series of other key thermogenic genes. Importantly, these Y1R signalling properties are conserved in humans with upregulation of Y1R expression found in both subcutaneous and visceral fat of obese individuals, showing a strong inverse relationship with the expression of thermogenic markers including UCP1 and PGC1α in these tissues. These results suggest that antagonism of Y1R signalling via BIBO3304 treatment releases its inhibitory tone on UCP1 and a number of other genes that control thermogenic pathways to maximize mitochondria activity leading to an increase in heat production. Moreover, body weight loss and fat loss are no longer observed in UCP1$^{-/-}$ mice treated with BIBO3304. Of note, although our data suggests that BIBO3304's effects on body weight and BAT function could be due to the blockade of Y1R on adipocytes, as Y1R is also important in vascular function[26], it is possible that Y1R's actions on non-adipose tissues including blood vessels may contribute.

Consistent with increased thermogenic activity in the mitochondria, lower RER, indicative of greater utilization of lipids as energy substrate in BIBO3304-treated mice especially under excess calorie conditions was observed. This is coupled with increased CPT1 expression in BAT, suggesting Y1R signalling controls mitochondrial capacity for fatty acid transport and lipid oxidation. Furthermore, increased phosphorylation of ACC by BIBO3304 treatment in BAT indicates an inhibition of lipid synthesis. Therefore, peripheral Y1R antagonism favours a catabolic rather than anabolic effect; greater lipid utilization and thermogenic capacity induced by Y1R antagonism may provide the mechanism leading to significantly reduced (e.g., smaller average adipocyte size) visceral fat (WATe and WATm) and thus less metabolically harmful intra-abdominal fat. Importantly, this fat reduction occurs in concert with a marked improvement in glucose tolerance, particularly under excess calorie conditions with BIBO3304 treatment. It is likely that BIBO3304-induced greater thermogenesis in BAT and browning of WATs utilizes and robustly clears lipids as well as glucose in the body. In addition, the induction of metabolically beneficial PPARγ and adiponectin expression upon peripheral Y1R antagonism may contribute to improved glucose tolerance. Furthermore, the real-time analysis of the dynamic changes in p-Akt activity in intrascapular BAT provides strong and direct evidence that peripheral antagonism of Y1R signalling improves insulin action in adipose tissues.

Notably, peripheral Y1R signalling activates distinct sets of thermogenic genes and pathways and demonstrates fat depot-specific expression profile under different nutritional states. Under a standard chow diet, BIBO3304 treatment generally leads to the upregulation of a number of beige signature genes such as CD137, Tmem26, and Klhl13 in WATi, without affecting the expression of classical brown fat genes UCP1, DIO2, CIDEA, Cox7a1 and PRDM16. This could explain, at least partly, the trend towards only slightly reduced body weight gain in chow-fed BIBO3304-treated mice. In contrast, consistent to previous studies, the expression of the majority of thermogenic markers was inhibited by HFD feeding; however, BIBO3304 is able to counteract the inhibitory effect of HFD on thermogenic programme and induce substantial elevation in a panel of thermogenic markers including UCP1, DIO2, CIDEA, Cox7a1, PGC1α, CD137 and Acot2 mRNA expression without altering the expression of Tmem26, Klhl13, Ebf3 and Slc27a1, resulting in a dramatic reduction in weight and fat loss. Similarly, p-CREB and p-ERK level were significantly upregulated in both BAT and WATi by BIBO3304 treatment under a chow diet. Conversely, under conditions of excess energy, the levels of p-CREB, but not p-ERK, was enhanced in BAT and both p-CREB and p-ERK levels were completely abolished in WATi. Thus, it is clear that both cAMP-CREB pathways and MAPK-ERK signalling pathways participate in mediating the effects of Y1R signalling in a diet- and fat depot- dependent manner. However, it can not be excluded that antagonism of Y1R signalling in non-adipose tissues, such as skeletal muscle, liver and bone, could also contribute to the overall metabolic benefits. Nevertheless, given that Y1R expression is mainly found in adipose tissue where it is also highly upregulated under positive energy balance conditions, which is not seen in other peripheral tissues, contributions from these non-adipose tissues are less likely. Further studies employing tissue-specific Y1R deletion mouse models are warranted to confirm this.

It is worth noting that Y1Rs are expressed in blood vessels where they act as vasoconstrictors alongside co-released noradrenaline[26]. Blockade of vascular Y1R will lead to vasodilation and reduced blood pressure providing an additional benefit. In addition, it is possible that altered blood flow could distribute heat generated in BAT and WAT more efficiently to tissues throughout the body including the tail where increased heat loss is detected.

Tamoxifen-induced Cre-mediated gene recombination is a well-established technology to precisely control target gene expression in a temporal and spatial manner avoiding the potential involvement of developmental compensation. However, tamoxifen administration can have effects on its own as seen by a considerable weight and fat mass increase in control littermate mice. The precise mechanisms for this phenomenon are not clear, but it could be due to a prolonged presence of tamoxifen, an oestrogen receptor ligand, in adipose tissues leading to adipocyte or preadipocyte proliferation. Notably, tamoxifen has been shown to be able to stimulate de novo adipogenesis[21]. Nevertheless, even on the background of this obesogenic effect of tamoxifen, deletion of the Y1R specifically in adipose tissues of Adipo-Y1$^{-/-}$ mice was able to normalize body weight and fat gain, highlighting the powerful fat-reducing effect of lack of Y1R signalling in adipose tissues. It remains unclear how Y1R deletion prevents tamoxifen-induced weight gain, but one of the possible explanation is that inhibition of Y1R signalling in adipocytes suppresses MAPK-mediated proliferation[33], thereby inhibiting tamoxifen-induced adipogenesis. It is also possible that adipocyte Y1R deletion-induced elevation in thermogenesis outstrips the tamoxifen-induced adipogenesis, leading to a net outcome of no change in body weight. Alternatively, tamoxifen-induced weight gain and Y1R deletion-induced normalization of body weight may be mediated by independent pathways. Further studies are warranted to test these possibilities. Of note, adipocyte-specific Y1R deletion decreases the body weight gain in both chow- and HFD-fed mice, whereas BIBO3304 was more effective in reducing the body weight gain in HFD-fed mice than in chow-fed mice.

Taken together, our results demonstrate both the critical role of peripheral Y1R signalling as a negative regulator of thermogenic activity in adipose tissues as well as how this could potentially be used for pharmacologically boosting thermogenesis to augment EE and reduce obesity. Critically, in contrast to conventional approaches that target brain-based pathways for the control of appetite and weight loss, our findings provide a fresh angle for obesity and diabetes intervention—periphery-based therapies. Considering that at least 80% of centrally acting anti-obesity medications have been withdrawn from the market due to severe side effects impacting cardiovascular health and psychiatric health, blocking Y1R signalling only in peripheral tissues provides an attractive, safer and potentially more effective therapeutic option for obesity and diabetes mitigation.

## Methods

**Animals**. All research and animal work were conducted under the regulation of Garvan Institute/St Vincent's Hospital Animal Ethics Committee and were in agreement with the Australian Code of Practice for the Care and Use of Animals for Scientific purpose. Mice were housed under conditions of controlled temperature (22 °C) and illumination (12-h light cycle, lights on at 07:00) with humidity ~50%. Age-matched male mice were fed on either a chow diet ad libitum (8% calories from fat, 21% calories from protein, 71% calorie from carbohydrate and 10.87 KJ/g, Gordon's Specialty Stock Feeds, Yanderra, NSW, Australia) or a high-fat diet (HFD, 43% calories from fat, 17% calories from protein, 20 KJ/kg; Gordon's Specialty Feeds, Glen Forrest, WA, Australia). Water was available ad libitum.

Age-matched male mice on a C57Bl/6JAusb background were used for all experiments unless stated otherwise. C57Bl/6JAusb mice were purchased from Australian BioResources Pty Ltd (ABR, Moss Vale, NSW, Australia). Generation of the Y1R$^{-/-}$, UCP1$^{-/-}$ and conditional Y1$^{lox/lox}$ mice has been described generated previously[9,34,35]. Generation of adiponectin-specific Y1R knockout (Adipo$^{TMCre/+}$; Y1$^{lox/lox}$) mice was described below.

Non-brain penetrable[12,36] specific Y1R antagonist BIBO3304 was purchased from Tocris Bioscience (BIBO3304 trifluoroacetate, #2412) and was dissolved in distilled H$_2$O at a concentration of 1 mg/mL, and aliquots were stored at −20 °C until required. To avoid the stress caused by gavage, BIBO3304 was incorporated into a jelly containing 4.9% wt/vol gelatine and 7.5% imitation chocolate flavouring essence and administrated to mice daily for the duration indicated in the text. This method of drug delivery was developed in our laboratory and described previously[11]. In brief, we trained mice to voluntarily consume a vehicle jelly prior

to the start of an experiment. After 2–5 days training, over 95% of mice ate the entire portion of jelly (195 μl for a 25 g mouse) within 1 min of being placed in the cage and maintained a high avidity for jelly throughout the study period. At the commencement of an experiment, mice received BIBO3304 containing jelly once per day for the time period stated in each study, while control mice received vehicle jelly.

**Generation of the inducible adiponectin-specific Y1R ablation mouse model**. Inducible adiponectin-Cre mice were obtained from the Jackson Laboratory (stock No 025124). Adiponectin-specific Y1R knockout (Adipo$^{TMCre/+}$;Y1$^{lox/lox}$) mice were generated by crossing Y1$^{lox/lox}$ mice[35] onto AdipoQ-Cre mice (C57BL/6-Tg (Adipoq-cre/ERT2)1Soff/J mice) where the Cre-recombinase gene is under the control of the adiponectin gene promoter. For tamoxifen treatment of AdipoQ-Cre, 4-hydroxytamoxifen (4-OHT, Sigma-Aldrich, St Louis, MO) was dissolved in a solution of 10% ethanol in sunflower seed oil at a concentration of 50 mg/mL. This adiponectin-activated Cre-mediated gene deletion was induced in mice at 6–7 weeks of age via gavage with 100 μl of this solution, three times at 2-day intervals. Mice were killed 2 weeks after the last dose to confirm the gene deletion. A subset of mice was undergone the similar tamoxifen-induced Cre recombination protocol followed with metabolic characterization using the same paradigm used in C57Bl/6JAusb and UCP1$^{-/-}$ mice. Commencement of induction of gene deletion was defined as the day of the first gavage.

**Human subcutaneous and visceral adipose tissue collection**. Volunteers were recruited from adults undergoing bariatric surgery (obese subjects) and routine abdominal surgery (lean subjects) at the Wuhan Union Hospital, Wuhan, China after ethical permission from the local research ethics committee (IORG No. 0003571). All the subjects involved in this study have given their informed consent, in accordance with the guidelines in the Declaration of Helsinki 2000. The participants were free from current diseases as determined by medical check-up and history. Exclusion criteria in this study were subjects with known diabetes, cardiovascular diseases, thyroid disease, renal impairment (serum creatinine > 120 μmol/l), hypertension (blood pressure > 140/90 mmHg) and malignancy. Subjects who were on medications that likely affected body weight and any study variables, for example, systemic glucocorticoids were also excluded from this study.

Obese group - 10 men and 9 women, aged 34.75 ± 8.86 years, BMI 42.06 ± 7.14 kg/m$^2$, $n = 19$; lean group – 6 men and 5 women, aged 39.72 ± 11.76 years, BMI 20.75 ± 1.98 kg/m$^2$, $n = 11$. Volunteers attended the clinical investigation unit for physical examination and venesection before admission for surgery. Blood pressure, weight and height were recorded by the same observer, and BMI was calculated. Basic characteristics of these subjects were shown in Supplementary Table 1.

Biopsies of visceral adipose tissue (VAT) around the stomach were obtained from participants undergoing elective abdominal laparoscopic surgery. Subcutaneous adipose tissue (SAT) biopsy samples were obtained near a trocar (a medical device that placed through the abdomen during laparoscopic surgery). Biopsies were either immediately frozen in liquid N$_2$ and stored at −80 °C for total RNA extraction, or kept in a warm cell culture medium and transported to the lab for SVF isolation and ex vivo culture as previously described[37]. Total RNA was extracted using Trizol Reagents (Sigma) and reversely transcribed for the determination of mRNA expression of members of the NPY family, thermogenic markers and lipogenic markers in SAT and VAT.

**Isolation of human SVF and culture ex vivo**. SAT biopsies from five obese subjects were collected (three men, two women; age 36.1 ± 6.5 years; BMI 39.4 ± 2.1 kg/m$^2$). SVF containing pre-adipocytes were isolated from the adipose tissue samples by 0.075% collagenase digestion (collagenase type 1, Worthington, NJ, # 4196) as described previously[37]. Adipocytes were cultured in a high glucose DMEM/F12 media (1:1) basal medium (Invitrogen) supplemented with 10% FBS, Glutamax and penicillin/streptomycin. When cells were confluent, cells were plated into 24-well plates, and added the differentiation cocktail containing basal medium supplemented with 5 μg/ml insulin, 1 μM dexamethasone and 0.5 mM iso-butylmethylxanthine. 1 μM rosiglitazone (PPARγ agonist) was added in the first 48 h of induction media to enhance differentiation. Differentiated adipocytes were then divided into four groups, and treated with Y1R-specific agonist $^{[Leu31,Pro34]}$ NPY (Tocris Bioscience, # 1176) and/or antagonist BIBO3304 for 48 h. Adipocytes were harvested for isolation of total RNA followed by qPCR for determination of thermogenic and lipogenic gene expression. Primer sequences used in qPCR analysis were in Supplementary Table 2.

**Determination of body weight and FI**. Male mice were housed individually throughout the whole experiment and body weight was monitored weekly over the experimental period. Mice were examined for FI, EE, RER, physical activity as described below and body composition at the times indicated in the text. Basal daily FI in the fed state was measured using a well-established protocol[38–40]. In brief, 7-week-old male C57Bl/6JAusb mice were individually housed and trained for voluntary jelly consumption, acclimatized for a soft paper bedding for 3 days and then fed a HFD with simultaneous daily administration of BIBO3304 or vehicle jelly. Spontaneous daily FI was measured from day 1 of treatment for

3 weeks during which a rapid DIO is generally observed in mice. Actual FI was calculated as the weight of pellets taken from the food hopper minus the weight of food spilled on the cage floor. The weight of spilled food per day was determined as the 24-h increase in weight of cage bedding, after removing all faeces and air-drying to eliminate weight changes due to urine. Cumulative FI over these 3 weeks was calculated. At week 2 and week 6 post treatment at 9 and 13 weeks of age, respectively, fasting-induced food intake (FIFI) was also measured as previously described[40]. Briefly, mice were weighed, individually housed, and fasted for 24 h. FI was subsequently measured at 2, 6, 24 and 48 h after refeeding with the respective types of diet, while the corresponding body weight was recorded in parallel.

**Body composition measurement and indirect calorimetry**. Body composition including whole-body fat mass and lean mass were measured at baseline, 2, 4 and 6 weeks post treatment or as indicated in each cohort using dual-energy X-ray absorptiometry (DEXA; Lunar PIXImus2 mouse densitometer, GE Medical Systems, Madison, WI) under light isoflurane anaesthesia as indicated in each cohort. The head and the tail were excluded from DEXA analysis. Oxygen consumption rate ($V_{O_2}$), carbon dioxide output ($V_{CO_2}$) and physical activity were measured at 12–13 weeks of age using an open circuit eight-chamber indirect calorimeter (Oxymax series; Columbus Instruments, Columbus, OH, USA) with airflow of 0.6 L/min and temperature maintained at 22 °C as previously described[4,38]. Briefly, mice were housed individually in metabolic chambers (20.1 × 10.1 × 12.7 cm). Mice were singly housed for 3 days before they were acclimatized to the metabolic chambers for 24 h prior to recordings commencement. Mice were monitored in the metabolic cages for 24 h. Oxygen consumption ($V_{O_2}$) and carbon dioxide production ($V_{CO_2}$) were measured every 27 min. ER was calculated as the quotient of $V_{CO_2}/V_{O_2}$, with 100% carbohydrate oxidation giving a value of 1 and pure fat oxidation giving a value of 0.7[41,42]. EE (kcal heat produced) and was calculated as caloric value (CV) × $V_{O_2}$, where CV is 3.815 + 1.232 × RER[43]. EE was normalized to metabolically active tissue (MAT) mass which is calculated as "lean mass + 0.2 × fat mass"[16,44]. Whole-body fat mass and lean mass were measured immediately following the completion of indirect calorimetry using DEXA. Physical activity was also measured within the metabolic cages using an OPTO-M3 sensor system where ambulatory counts were a record of consecutive adjacent photo beam breaks in the horizontal space (X and Y directions). Data for the 24 h monitoring period was presented as hourly averages for $V_{O_2}$, $V_{CO_2}$, RER, and EE as well as hourly summation for locomotor activities. The calorimeter was calibrated before each use using highly pure primary gas standards ($O_2$ and $CO_2$).

**Body, BAT and tail temperature measurements using infra-red imaging**. The temperature of interscapular BAT ($T_{BAT}$) and body temperature ($T_{lumbar\ back}$) was measured by non-invasive high-sensitivity infra-red imaging[4,45] at the times indicated in each cohort. Briefly, male mice had their interscapular and lumbar back skin shaved under light isoflurane anaesthesia one day prior to experiments. All animals had been acclimatized in the testing cages for 3 days before measurement. A high-sensitivity infra-red camera (ThermoCAM T640, FLIR, Danderyd, Sweden, sensitivity = 0.04 °C) fixed on a tripod was placed 90 cm above the freely moving single-housed mouse, and thermographic images were taken for 1 min per mouse. After each 1-min measurement, the camera was moved to the next animal. Upon completion of the first round of measurement, a second round and third round were conducted. The readings from the three rounds of measurements were averaged to represent the temperature in that day. Temperature measurement was conducted once a day for 2 consecutive days at the same time of the day under ambient temperature conditions (22 °C). The temperature readings of the 2 days were averaged. The videos were analyzed with the ThermaCAM 7 Pro software (FLIR ResearchIR Max, version 4.40.8.28) and the thermo frames that had mouse body naturally extended with both shaved skin regions vertical to camera lenes were extracted from the videos. The hottest pixels of the lumbar back, interscapular regions and tail (1 cm from the base of tail) from each extracted frame were obtained simultaneously and used as indicators of body, BAT and tail temperature, respectively. The average temperature of lumbar back, intrascapular BAT and tail was calculated.

**Glucose and insulin tolerance tests (GTT and ITT)**. Mice were fasted for 6 h before intraperitoneal (i.p.) injection of a 10% D-glucose solution (1.0 g/kg body weight)[38,46] at age indicated in the main text. Blood samples were obtained from the tail tip at the indicated times, and glucose levels were measured using a glucometer (AccuCheck II; Roche, New South Wales, Castle Hill, Australia). Serum from mice administrated with glucose was collected and stored for subsequent insulin assay using a Rat/Mouse insulin RIA kits (Linco Research, St Charles, MO, USA). Serum insulin levels at fed and fasted states of these mice were also determined using the same insulin RIA kits following the manufacturers' instructions. ITT was carried out to determine insulin-induced hypoglycemia in mice at ages indicated in the main text. Briefly, the mice were injected with insulin (1 IU/kg body weight) intraperitoneally to induce hypoglycemia after being fasted for 6 h, then blood samples were collected from the tail tip at the indicated times, glucose levels were measured using a glucometer (Roche) as shown in the results.

**Implantation of optical imaging windows**. Mice constitutively expressing the Eevee-Akt-FRET biosensor were engrafted with titanium mammary imaging windows (Russell Symes & Company) as previously described[24,25] over intrascapular BAT. Briefly, mice were treated with 5 mg/kg of the analgesic Carprofen (Rimadyl) in pH neutral drinking water 24 h prior and up to a minimum of 72 h post surgery. Mice further received subcutaneous injections of buprenorphine (0.075 mg/kg, Temgesic) immediately prior and 6 h post surgery. The titanium window was prepared 24 h prior to surgery by gluing a 12 mm glass coverslip (Electron Microscopy Science) using cyanoacrylate to the groove on the outer rim of the titanium window. Following anaesthetic induction at 4% isoflurane delivered via a vaporizer (VetFlo) supplemented with oxygen, mice were kept at a steady 1–2% maintenance anaesthesia for the duration of the surgery on a heated pad. The incision site was shaved and depilated (Nair) and disinfected using 0.5% chlorhexidine/70% ethanol. A straight incision was made into the skin above the intrascapular BAT and following blunt dissection of the skin surrounding the incision a purse string suture (5–0 Merslik, Ethicon) placed. The windows were then inserted and held in place by tightening the suture, disappearing along with the skin into the groove of the window and tied off. Mice were allowed to recover for a minimum of 72 h post surgery, actively foraging, feeding and grooming within minutes from being removed from the anaesthesia respirator. A minimum of 24 h prior to imaging mice was weaned off the Carprofen analgesic in the drinking water.

**In vivo imaging**. Mice were imaged under 1–2% isofluorane on a heated stage (Digital Pixel, UK) prior to and after i.p. injection of a 10% D-glucose solution (1.0 g/kg body weight). Multiphoton imaging was performed using a Leica DMI 6000 SP8 confocal microscope using a 25 × 0.95 NA water immersion objective on an inverted stage. The Ti:Sapphire femto-second laser (Coherent Chameleon Ultra II, Coherent) excitation source operating at 80 MHz was tuned to a pumping wavelength of 840 nm. RLD-HyD detectors were used with 435/40 and 483/40 bandpass emission filters to detect the second harmonic generation (SHG) of the collagen I and mTurquoise2, respectively. The imaging depth was about 60 μm beneath the window. Images were acquired at a line rate of 700 Hz, 512 × 512 pixel and at a total of 203 frames per image. Realignment of the data was performed using Galene (v2.0.2)[47] using the warp realignment mode, 10 realignment points, a smoothing radius of 2 px and a realignment threshold of 0.4 applied for the SHG channel and 0.6 for the mTurquoise2 signal.

**RNA extraction and quantitative real-time PCR**. Different tissues including WAT and BAT were dissected and immediately frozen in liquid nitrogen and total RNA was extracted using Trizol Reagent (Sigma, St. Louis, MO). The quality and concentration of total RNA was measured by a spectrophotometer (Nanodrop 1000, NanoDrop Technologies, LLC, USA). 1 μg total RNA was reversely transcribed using a Superscript III First-strand cDNA Synthesis Kit (Invitrogen, Mount Waverley, VIC, Australia). Real-time RT-PCR was performed as previously described[48] using a LightCycler® (Light-Cycler 480 Real-time PCR System, Roche Applied Science, Germany). The value obtained for each gene product was normalized by the ΔΔCT method to a housekeeping gene ribosomal protein L19 (Rpl19) in mouse samples or β-actin in human samples and expressed as a fold change of the value obtained for the controls. Primer sequences are listed in Supplementary Table 3. The amplification condition used in all the RT-PCR experiments was: 94 °C for 30 s, 60 °C for 30 s, 72 °C for 20 s for 40 cycles unless stated otherwise.

**Immunohistochemical staining**. UCP1 immunohistochemistry was performed on paraformaldehyde (PFA)-fixed, paraffin-embedded adipose tissue sections. Briefly, WAT from control and treatment group was fixed in 4% PBS-buffered PFA overnight at 4 °C before being processed and embedded in paraffin. Sections were cut at 5 μm, deparaffinised and rehydrated. Slides were then washed in distilled water and incubated with 5% goat serum in 1× PBS containing 2% BSA for 1 h at room temperature. Subsequently sections were incubated overnight at 4 °C in hydration chambers with anti-mouse UCP1 primary antibody (Alpha Diagnostics, produced in rabbit, 1:1000 dilution). Slides were then washed in 1× PBS and incubated with the biotinylated secondary antibody (Vector Lab, produced in goat, #BA1000, 1:500 dilution) for 1 h at room temperature. The slides were washed in PBS and incubated with Extravidine (Sigma-Aldrich, #E2886, 1:250 dilution) in 1× PBS for 30 min followed by wash with 1× PBS. The sections were incubated in peroxidase substrate solution until desired stain intensity developed. Slides were coverslipped with mounting medium. Sections were imaged using a Zeiss Axioplan light microscope equipped with the ProgRes digital camera (Carl Zeiss Imaging Solutions, GmbH, Munich, Germany).

**Western blotting analysis**. WAT and BAT samples from all groups were homogenized in RIPA buffer as previously described[39], and 20 μg protein was resolved by SDS-PAGE and immunoblotted with antibodies against UCP1 (Alpha Diagnostic International, San Antonio, TX, #UCP11-A, 1:1000 dilution), PGC1α (Millipore, #AB3242, 1:1000 dilution), CPT1 (Alpha Diagnostic International, #CPT1M11-A, San Antonio, TX, 1:1000 dilution), Mitomix (Abcam, #MS604, 1:1000 dilution), p-CREB (Cell Signalling Technology, #9198, 1:1000 dilution), p-

ACC (Cell Signalling Technology, #3661, 1:1000 dilution), p-ERK (Cell Signalling Technology, #9102, 1:1000 dilution) and 14-3-3 (Santa Cruz, #Sc629, 1:1000 dilution). Secondary antibodies, donkey anti-rabbit IgG-HRP (H + L) (Jackson ImmunoResearch, #711-035-152) and sheep anti-mouse IgG-HRP linked (GE Healthcare, #NA931, 1:5000 dilution) were used. Immunolabelled bands were quantified by densitometry using ImageJ software (1.38×, National Institutes of Health, USA).

**Tissue collection and analysis.** At the completion of the study, mice were killed by cervical dislocation between 12:00 and 15:00, trunk blood was collected, allowed to clot at room temperature, centrifuged, and the resultant serum was stored at −20 °C for subsequent analysis. WAT depots (right inguinal, right retroperitoneal, right epididymal, and mesenteric), and BAT were collected, weighed and stored at −80 °C for further analysis.

**Determination of adipocyte size and number.** For the histopathological analysis WATi and WATe were instantly dissected out and fixed in 4% neutral buffered formalin solution. After routine processing, the adipose tissues were embedded in paraffin and cut at a thickness of 5 μm. Sections were stained with H&E, and subsequently examined using a light microscope for histopathological examination. The entire WAT sections were scanned using a Power Mosaic microscope (Leica), examined and quantified by an independent trained researcher. In brief, quanti-fication was carried out on a Leica DM6000 microscope equipped with a HC PLAN APO 10× (0.4) and 20× (0.7) objectives, a DFC310FX camera and LAS "Power Mosaic" software. The entire WAT sections were analyzed using ImageJ/FIJI software[49]. Images were smoothed and simplified using colour thresholds to allow elimination of non-adipose tissue and identification and size/shape analysis of regions/particles representing adipocytes. Colour thresholding to identify likely adipocytes was performed in the HSB colour space (H 0–255; S 38–255, B 50–255) and subsequently identified particles (100–25,000 px, circularity 0.375–1.000) were scored for area, roundness and solidity and results analyzed using Microsoft Excel.

**Statistical analyses.** All values are presented as mean ± s.e.m. EE, FI, RER and physical activity over the continuous 24 h period were averaged for the whole 24 h period, as well as for the light and dark period. One- or two-way ANOVA, repeated measures ANOVA, or Student's $t$ test (two-tailed) was used to determine the significance of treatment effects and interactions (GraphPad Prism 9 for MacOS, version 9.0.0; GraphPad Software, Inc). When there was a significant overall effect or interaction effect, Tukey's or Sidak's multiple comparisons tests were performed to identify differences among means. For all statistical analyses, $p < 0.05$ was regarded as significant.

**Reporting summary.** Further information on research design is available in the Nature Research Reporting Summary linked to this article.

## Data availability
The data supporting the findings of this study are available from the corresponding authors upon reasonable request. Source data are provided with this paper.

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

## Acknowledgements

We are grateful for the technical assistance from Ireni Clark, Felicia Reed, Gopana Gopalasingam, Xuan Zhang and Alexander Zinn. We also thank Michael Pickering and Keith Cheung of the Garvan Institute Biologcal Testing Facility for facilitation of the animal experiments. This research was supported by National Health & Medical Research Council (NH&MRC) grant #1144286, Diabetes Australia Research Programme (DART) grants (#Y17G-ShiY and #Y19G-ShiY) to Y.C.S. and NH&MRC fellowship #1118775 and NH&MRC grant #1123877 to H.H.

## Author contributions

Y.C.S. conceived the idea, managed the project, designed and performed experiments, analyzed data, wrote manuscript, and reviewed/edited manuscript. H.H. conceived the idea, contributed critical discussion, and reviewed/edited manuscript. C.X.Y., K. L. performed in vivo animal experiments & analyzed phenotypic data, conducted qPCR, and performed immunohistochemical & western blotting analyses. T.S.Z. contributed conceptualization of experiments of human adipose tissues, managed, performed ex vivo human experiments, cell culture, conducted qPCR and data analysis along with L.N.G. Y.C.S. and M.N. designed and performed studies in Akt-FRET biosensor mice, and M.N. performed surgical window implantation, imaged, analyzed and interpreted data arising from the Akt-FRET mice. K. L. contributed to histological examination, interpretation and result discussion. Z.F.X. performed human surgery and provided clinical adipose tissue samples. Z.M.G., J.L., R.E., H.Y.G. and Q.W. performed in vivo animal experiments and radioimmunoassays. M.B. performed histological examination and contributed discussion. C.K.I., L.Z., and Q.P.W. contributed discussion. W.H. analyzed adipocyte size and distribution. D.R.L. contributed discussion. J.J.H. and P.T. generated and provided Akt-FRET biosensor mice. All authors read and approved the final manuscript.

## Competing interests

The authors declare no competing interests.
