## [Peer Review File · Nature Communications]

Reviewer comments, first round –

Reviewer #1 (Remarks to the Author):

Review of NCOMMS-20-32677-T BIBO3304, a non-brain penetrable Y1 receptor antagonist increases 1 thermogenesis and protects against diet-induced obesity

This comprehensive paper describes the physiological effects of non-brain penetrant NPY Y1 R antagonist BIBO3304 on a variety of energy homeostasis parameters, including weight gain, glucose tolerance, insulin tolerance, energy expenditure, white and brown fat trans differentiation and intracellular signalling.

Initially studies show that there is differential expression of NPY Y1 R in various depots in fat from lean vs obese mice. A similar effect was observed in humans. The paper then clearly demonstrates that Y1 blockade exert metabolically beneficial effects on body weight (reduced gain) and glucose tolerance (increased). It does this by increasing EE in BAT and other fats, by increased expression of "brown" gene in WAT. Both factors increases BAT temperature and also tail temperature. They also show that induced deletion of Y1R mirrors the effects of acute and chronic antagonism of Y1R.

The data package supports further evaluation of BIBO3304 and other NPY Y1 R antagonists for metabolic disease.

The studies are well conducted and well described. The data are believable. Remarkably with such a large data set the studies all find complimentary results, adding power to the conclusion that are reached. The data are well presented

The introduction sets up the premise for the experiments. And the discussion is appropriate.

It is pleasing to see human data included in the studies.

This is an excellent paper.

Reviewer #2 (Remarks to the Author):

The manuscript by Yan et al. described the role of peripheral Y1R signaling for controlling energy expenditure and reduction of obesity. The authors analyzed Y-receptor expression level in various tissues including white fat tissue and brown adipose tissue in mice under chow and high fat feeding conditions, which indicates upregulation of Y1R expression in adipose tissues. Then they evaluated the effects of Y1R antagonism on diet-induced obesity using a selective Y1R antagonist BIBO3304, which revealed that the Y1R antagonism significantly reduced the mice body weight. Y1R antagonism also induced enhanced thermogenesis in brown adipose tissue and browning of white adipose tissue with the alteration of the gene expression pattern. In addition, treatment with Y1R antagonist induced cAMP-CREB and MAPK-ERK signaling pathways and upregulated glucose-induced metabolic signaling.

This study demonstrated the involvement of Y1R signaling in energy expenditure in adipose tissues and reduction of body weights. The authors did considerable efforts and the presented data are well discussed to support the conclusion. However, I have some concerns about the data (mainly in multiphoton imaging data) in the manuscript. After addressing them, I recommend that this manuscript will be acceptable for publication in Nature Communications.

In Figure 11f, the dynamic changes of Akt phosphorylation was visualized using an Akt-FRET biosensor in living mice. From the imaging results, the authors mentioned that the Akt activity is enhanced and maintained throughout 30-minute period after glucose administration. However, I think that this might be overstated. The imaging data showed the decrease of the lifetime (enhancement of Akt activity) after 2-minutes injection; however, the Akt activity is then recovered to the original level. The difference of the activity between vehicle and drug-treated mice is not significance from 10-30 minutes.

The authors performed multiphoton imaging to analyze Akt phosphorylation dynamics in living

mice using a FRET-based fluorescent probe. Considering the presented imaging data in Figure 11f, the authors should clearly describe that fluorescence lifetime imaging was performed in this section. Also, the unit (ns) should be added in the scale in Figure 11f.

Did the authors validate the specificity of the FRET probe? I think that Imaging experiment with Akt inhibitor would validate the specificity of the FRET probe and support the conclusion.

The method for the adipocyte size analysis (Figure 7) and the information of the imaging depth in multiphoton imaging experiment (Figure 11f) should be described in the experimental section.

In Figure 11b and d, the legend for the graph should be added.

Reviewer #3 (Remarks to the Author):

This paper examines the ability of Neuropeptide Y (NPY) type 1 receptors (Y1R) expressed in brown and white adipose tissue (BAT and WAT) to counteract obesity in mice that are lean or made obese via intake of a high fat diet (HFD). A key aspect of the approach is the use of BIBO3304, which the authors suggest cannot cross the blood-brain barrier (BBB) and thus is effective only in the periphery. They show that BAT and WAT are particularly enriched with Y1R in mice and also humans (WAT only examined). A HFD in mice and obesity in humans upregulate Y1R. Conversely, chronic blockade of Y1R with BIBO3304 or adipose-specific deletion of Y1R counteracts diet-induced obesity (DIO) by increasing energy expenditure, activating BAT (increased temperature and UCP-1), and inducing WAT browning (increased UCP-1). Collectively, the authors conclude that BIBO3304 could be developed for use as a safer and more effective treatment to counteract DIO.

The paper has many strengths, in particular a novel investigation of the roles and actions of BAT and WAT Y1R as well as the multiple approaches used to investigate Y1R significance. The ability of NPY to decrease UCP-1, and BIBO3304 to increase UCP-1, in human adipocytes in culture is strong evidence that NPY via Y1R can directly inhibit WAT metabolism. However, a major concern is that the effects of Y1R blockade or deletion are not mediated exclusively by Y1R expressed in adipocytes, which contradicts the main conclusions.

MAJOR SPECIFIC COMMENTS.

1. The authors explicitly state that BIBO3304 acts only in the periphery; however, the papers cited to support this contention do not appear to directly assess this. Moreover, another study demonstrated that chronic peripheral administration of BIBO3304 in mice decreases food intake, suggesting brain access, although interestingly in rats BIBO3304 was effective only when combined with a Y5 antagonist (PMID: 10737680). The authors use the failure of BIBO3304 to inhibit food intake in mice as further evidence for its inability to cross the blood-brain barrier. However, the food intake measurements were made only on one day and after body weights had stabilized (i.e. no further increments in BW). The effects of BIBO3304 on energy expenditure at this time are also quite modest. To validate these findings, more frequent and/or prolonged measurements of cumulative food intake should be conducted during DIO development. Alternatively, pharmacokinetic measurements of BIBO3304 (i.e. whether it crosses the BBB) could be performed. If BIBO3304 can access the brain, then many of the findings can be explained by reversal of NPY's chronic central action to suppress BAT activity.

2. Besides CNS effects, at least some of the effects of BIBO3304, such as on BAT temperature, may be mediated by blockade of vascular Y1R in BAT or WAT as in the tail. In this context, the associated effects of BIBO3304 on cardiovascular function and blood pressure need to be considered.

3. The ability of UCP-1 deletion to inhibit the effects of BIBO3304 on BW and BAT may not be specific to direct effects of Y1R on adipocytes. Loss of UCP-1 could prevent other more indirect effects as well, such as CNS or vascular actions. The authors should analyze Figure 8 d and 8e as in Figure 1.

4. A key experiment was determining if tamoxifen-induced loss of the Y1R in adult mice selectively in adipocytes mimics the effects of BIBO3304. However, the results are confounded by the effects of tamoxifen alone to increase BW in mice eating chow or a HFD. It is difficult to understand how Y1R deletion prevents tamoxifen-induced weight gain, since the mechanism for this gain is unknown. It is also confounding that adipocyte Y1 deletion decreases this body weight gain in both chow- and HFD-fed mice, when BIBO3304 was effective only in HFD mice. This result requires further explanation and possibly experimentation.

5. There are some inconsistencies that should be addressed or discussed: 1) BIBO3304 alters WAT_i versus WAT_i indices (e.g. weight, Y1R expression, size) variably. 2) A HFD/obesity increases UCP-1 in mice but decreases UCP-1 in humans.

6. The paper is descriptively written. It would benefit from a testable mechanistic hypothesis in the Introduction. In addition, the discussion is largely a restatement of the results, and the main conclusion is vague and more related to the potential use of BIBO3304 to counteract obesity. I recommend that the results and their discussion are built more like a story with mechanistic conclusions at each step. Instead, the discussion could include a cartoon that summarizes the findings and discusses in more detail the potential physiological and pathophysiological implications. For example, where does the NPY originate that binds BAT or WAT Y1R? How is this NPY regulated physiologically or pathophysiology?

MINOR SPECIFIC COMMENTS.

1. Methodological concerns: How was food intake measured? How were mouse lumbar and BAT temperatures measured if the mice were unrestrained? Should energy expenditure be normalized to body weight, as more fat would increase energy expenditure independently of metabolism or activity level. Do differences in adipocyte size (or fat content) contribute to the PCR/Western results? What is the source of NPY in the cultures?

2. Acknowledge that some of the data are confirmatory, e.g. p 6 and 8. PMID 23838112 and PMID 19918245.

3. Fig 10c. But the suppression of UCP-1 by NPY was not reversed by BIBO3304?

RESPONSES TO REVIEWER COMMENTS

Reviewer #1 (Remarks to the Author):

Review of NCOMMS-20-32677-T “BIBO3304, a non-brain penetrable Y1 receptor antagonist increases 1 thermogenesis and protects against diet-induced obesity”

This comprehensive paper describes the physiological effects of non-brain penetrant NPY Y1 R antagonist BIBO3304 on a variety of energy homeostasis parameters, including weight gain, glucose tolerance, insulin tolerance, energy expenditure, white and brown fat trans differentiation and intracellular signalling.

Initially studies show that there is differential expression of NPY Y1 R in various depots in fat from lean vs obese mice. A similar effect was observed in humans. The paper then clearly demonstrates that Y1 blockade exert metabolically beneficial effects on body weight (reduced gain) and glucose tolerance (increased). It does this by increasing EE in BAT and other fats, by increased expression of “brown” gene in WAT. Both factors increase BAT temperature and also tail temperature.

They also show that induced deletion of Y1R mirrors the effects of acute and chronic antagonism of Y1R.

The data package supports further evaluation of BIBO3304 and other NPY Y1 R antagonists for metabolic disease.

The studies are well conducted and well described. The data are believable. Remarkably with such a large data set the studies all find complimentary results, adding power to the conclusion that are reached. The data are well presented

The introduction sets up the premise for the experiments. And the discussion is appropriate. It is pleasing to see human data included in the studies.

This is an excellent paper.

Our response: We appreciate the reviewer’s positive and encouraging comments on our study.

Reviewer #2 (Remarks to the Author):

The manuscript by Yan et al. described the role of peripheral Y1R signaling for controlling energy expenditure and reduction of obesity. The authors analyzed Y-receptor expression level in various tissues including white fat tissue and brown adipose tissue in mice under chow and high fat feeding conditions, which indicates upregulation of Y1R expression in adipose tissues. Then they evaluated the effects of Y1R antagonism on diet-induced obesity using a selective Y1R antagonist BIBO3304, which revealed that the Y1R antagonism significantly reduced the mice body weight. Y1R antagonism also induced enhanced thermogenesis in brown adipose tissue and browning of white adipose tissue with the alteration of the gene expression pattern. In addition, treatment with Y1R antagonist induced cAMP-CREB and MAPK-ERK signaling pathways and upregulated glucose-induced metabolic signalling.

This study demonstrated the involvement of Y1R signalling in energy expenditure in adipose tissues and reduction of body weights. The authors did considerable efforts and the presented data are well discussed to support the conclusion. However, I have some concerns about the data (mainly in multiphoton imaging data) in the manuscript. After addressing them, I

recommend that this manuscript will be acceptable for publication in Nature Communications.

Our response: Thank you for the positive comments.

In Figure 11f, the dynamic changes of Akt phosphorylation was visualized using an Akt-FRET biosensor in living mice. From the imaging results, the authors mentioned that the Akt activity is enhanced and maintained throughout 30-minute period after glucose administration. However, I think that this might be overstated. The imaging data showed the decrease of the lifetime (enhancement of Akt activity) after 2-minutes injection; however, the Akt activity is then recovered to the original level. The difference of the activity between vehicle and drug-treated mice is not significance from 10-30 minutes.

Our response: Thank you for the helpful comment. The phosphorylation of Akt has been used as a cellular marker to assess insulin sensitivity in insulin-responsive tissues including brown adipose tissue (Czech, 2020). Upon glucose administration, Akt phosphorylation occurs within 30 seconds in adipose tissue (Tan et al., 2015) and persists for up to 30 minutes. Impaired Akt phosphorylation in insulin signalling in adipocytes has been well-documented under conditions of high fat diet (HFD) and obesity (Chavez et al., 2003, Teruel et al., 2001, Huang et al., 2018). We observed significant enhancement of Akt activity at 2 minutes post a glucose bolus and an apparent but non-significant trend towards an increase in Akt activity in the subsequent time points up to 30 minutes as shown in the original **Fig. 11f**. This persistent trend of elevated Akt activity in response to glucose challenge in 4-week high fat diet fed BIBO3304-treated mice signifies the improved insulin sensitivity, which likely contributed to a better whole-body glucose tolerance observed in HFD-fed BIBO3304-treated mice. We have amended the description of the non-significant increase in Akt activity at the later time period in Results (on **Page 19**) of our revised manuscript.

The authors performed multiphoton imaging to analyze Akt phosphorylation dynamics in living mice using a FRET-based fluorescent probe. Considering the presented imaging data in Figure 11f, the authors should clearly describe that fluorescence lifetime imaging was performed in this section. Also, the unit (ns) should be added in the scale in Figure 11f.

Our response: We have added a more detailed description of how the fluorescence lifetime imaging was performed by multiphoton microscopy in Results on **Page 18-19**. The unit (nanosecond, ns) has also been added in the updated version of **Figure 11f** in the revised manuscript.

Did the authors validate the specificity of the FRET probe? I think that Imaging experiment with Akt inhibitor would validate the specificity of the FRET probe and support the conclusion.

Our response: Thank you for the helpful comment. The Akt-FRET biosensor has been validated previously (Komatsu et al., 2011) and the biosensor containing the mTurquoise2-YPet FRET pair has further been used previously to explore induction of Akt signalling during hypoxia in pancreatic cancer (Conway et al., 2018). In this paper we used an AKT pathway inhibitor, AZD2014, which inhibits both mTOR1/2, to inactivate Akt in pancreatic tumours in

conjunction with the hypoxia inducible drug TH-302. We have now added an additional experiment using the same inhibitor to demonstrate that Akt can be effectively inhibited in brown adipose tissue of the Akt-FRET biosensor mice (**Reviewer Figure 1**). This new data is now included as **Supplementary Fig. 6**.

Reviewer Figure 1: Brown adipose tissue (BAT) isolated from chow-fed Akt-FRET biosensor mice were treated *ex vivo* with 500 nM AZD2014 for 30 minutes and Akt activity quantified, revealing effective inhibition of Akt in this tissue. Values are mean \pm s.e.m. from 3 mice per group, 150 cells; * $P < 0.05$ unpaired students *t* test (two-tailed).

The method for the adipocyte size analysis (Figure 7) and the information of the imaging depth in multiphoton imaging experiment (Figure 11f) should be described in the experimental section.

In Figure 11b and d, the legend for the graph should be added.

Our responses: As described in our original manuscript, we used H&E staining of adipocyte plasma membranes to determine the size of adipocytes followed by automated measurement using Image J/FIJI software. We have now provided a more detailed description of the method for our adipocyte size measurement and included this reference (Schindelin et al., 2012) on **Page 34** of our revised manuscript.

The information of the multiphoton imaging depth (~60 μm beneath the optical window) has been added in the Methods (on **Page 32**) of our revised manuscript.

The figure legend for **figure 11b** and **11d** has also been added on **Page 47** in the revised manuscript.

Reviewer #3 (Remarks to the Author):

This paper examines the ability of Neuropeptide Y (NPY) type 1 receptors (Y1R) expressed in brown and white adipose tissue (BAT and WAT) to counteract obesity in mice that are lean or made obese via intake of a high fat diet (HFD). A key aspect of the approach is the use of BIBO3304, which the authors suggest cannot cross the blood-brain barrier (BBB) and thus is effective only in the periphery. They show that BAT and WAT are particularly enriched with Y1R in mice and also humans (WAT only examined). A HFD in mice and obesity in humans upregulate Y1R. Conversely, chronic blockade of Y1R with BIBO3304 or adipose-specific deletion of Y1R counteracts diet-induced obesity (DIO) by increasing energy expenditure, activating BAT (increased temperature and UCP-1), and inducing WAT browning (increased

UCP-1). Collectively, the authors conclude that BIBO3304 could be developed for use as a safer and more effective treatment to counteract DIO.

The paper has many strengths, in particular a novel investigation of the roles and actions of BAT and WAT Y1R as well as the multiple approaches used to investigate Y1R significance. The ability of NPY to decrease UCP-1, and BIBO3304 to increase UCP-1, in human adipocytes in culture is strong evidence that NPY via Y1R can directly inhibit WAT metabolism. However, a major concern is that the effects of Y1R blockade or deletion are not mediated exclusively by Y1R expressed in adipocytes, which contradicts the main conclusions.

Our response: Thank you for acknowledging that our study has many strengths. Based on our data that specific deletion of Y1R specific in adipocytes largely recapitulate the metabolic phenotypes observed in BIBO3304-treated wild type mice, we suggest that the Y1R in the brown and white adipocytes is the most likely major target for BIBO3304's metabolic benefits. However, as in our original manuscript, we acknowledge that Y1Rs are also expressed in other peripheral tissues including skeletal muscle, liver and bone, though to a far lower degree than in adipose tissue, and as such while inhibition of Y1Rs in these tissues by BIBO3304 could potentially also contribute to the overall beneficial effects, this may only represent a minor component. We have made this clearer now in the Discussion (on **Page 23**) of our revised manuscript. Further studies are warranted to confirm the contribution from other Y1R-expressing tissues, yet this is outside the scope of this manuscript.

MAJOR SPECIFIC COMMENTS.

1. The authors explicitly state that BIBO3304 acts only in the periphery; however, the papers cited to support this contention do not appear to directly assess this. Moreover, another study demonstrated that chronic peripheral administration of BIBO3304 in mice decreases food intake, suggesting brain access, although interestingly in rats BIBO3304 was effective only when combined with a Y5 antagonist (PMID: 10737680). The authors use the failure of BIBO3304 to inhibit food intake in mice as further evidence for its inability to cross the blood-brain barrier. However, the food intake measurements were made only on one day and after body weights had stabilized (i.e. no further increments in BW). The effects of BIBO3304 on energy expenditure at this time are also quite modest. To validate these findings, more frequent and/or prolonged measurements of cumulative food intake should be conducted during DIO development. Alternatively, pharmacokinetic measurements of BIBO3304 (i.e. whether it crosses the BBB) could be performed. If BIBO3304 can access the brain, then many of the findings can be explained by reversal of NPY's chronic central action to suppress BAT activity.

Our responses: We appreciate the reviewer's constructive comments. Promotion of feeding on a negative energy balance (i.e. fasting) is the hallmark of the activation of hypothalamic Arc NPY neurons, and this action is primarily mediated by the Y1/Y5 receptors (Nguyen et al., 2012). Importantly, intracerebroventricular (ICV) injection of the highly selective Y1R antagonist BIBO3304 has been shown to strongly suppress NPY-induced feeding (Wieland et al., 1998), highlighting the crucial role of central Y1R in the control of feeding.

In our original manuscript, we have performed spontaneous food intake measurements during the dark phase when mice are more actively eating, the light phase and over a 24-hour period

in the first week of treatment (aged 8 weeks). We also performed fasting-induced food intake measurements in the 2nd week (aged 9 weeks) as well as 5th-6th week of treatment (aged 13-14 weeks), when a big increment in body weight of control-treated mice was observed (**Fig. 1f-g**). These experiments demonstrated BIBO3304 treatment does not alter food intake under both fed and fasted states. Nevertheless, to further confirm the peripheral specific nature of BIBO3304, we have conducted more frequent and prolonged measurements of food intake.

7-week-old male C57BL/6J mice were individually housed, acclimatized for a paper bedding for 3 days and then fed a chow or HFD with simultaneous daily administration of BIBO3304 or vehicle jelly. Spontaneous daily food intake was measured in the first 3 weeks (aged 8-10 weeks) during which a rapid diet-induced obesity is generally observed in mice (Winzell and Ahren, 2004). Actual food intake was calculated as the weight of pellets taken from the food hopper minus the weight of food spilled on the cage floor as described previously (Shi et al., 2013, Ip et al., 2019). Briefly, the weight of spilled food per day was determined as the 24-h increase in weight of cage bedding, after removing all faeces and air-drying to eliminate weight changes due to urine. Our additional results show that both periodic and cumulative food intake is indistinguishable between BIBO3304-treated and control-treated mice on either chow or high fat diets over this 3-week period (**Reviewer Figure 2A-D**). We also measured spontaneous food intake on Week 5 post treatment (aged 13 weeks) where again daily food intake did not differ between treatment in either diets (**Reviewer Figure 2E**). Taken together, these results provide compelling evidence that BIBO3304 does not disrupt central Y1R signalling which would lead to a change in food intake. These data further confirm that the observed metabolic benefits of BIBO3304 are due to peripheral antagonism of Y1R signalling, excluding the involvement of central Y1R signalling in this regulatory process. This set of food intake data has now been added to our revised manuscript in **new Fig. 2a-b and new Supplementary Fig. 2a-c**. The methods of spontaneous and fast-induced food intake have been described in more detail (on **Page 28**) in our revised manuscript.

Reviewer Figure 2: Spontaneous daily food intake measured over first 3 weeks and at week 5 post treatments. (A) Daily food intake, (B) average daily food intake in the first 3 days of treatment, (C) cumulative food intake in the first 3 days, (D) between day 6-20 days and (E) at Week 5 post treatment of 7-week-old C57BL/6J male mice were fed a chow or HFD and simultaneously administered either control- or BIBO3304-containing jelly daily for 5 weeks. Values are mean \pm s.e.m. of 8 mice per treatment in HFD groups and 5 mice per treatment in chow groups. $P > 0.05$ within the same diet determined by two-way ANOVA analysis.

2. Besides CNS effects, at least some of the effects of BIBO3304, such as on BAT temperature, may be mediated by blockade of vascular Y1R in BAT or WAT as in the tail. In this context, the associated effects of BIBO3304 on cardiovascular function and blood pressure need to be considered.

Our responses: Thank you for the comments. We agree with the reviewer that the NPY system in particular Y1R is very important in vasoconstriction (Hirsch and Zukowska, 2012); and it cannot be excluded that some of the effects of BIBO3304 on BAT temperature may be mediated by blockade of vascular Y1R in BAT or WAT. Blocking vascular Y1R results in

vasodilation that possibly contributes to the elevated BAT temperature as well as whole body temperature. An increase in blood flow could also distribute heat from BAT and beige adipocytes throughout the whole body including the tail where increased heat loss was detected. This aspect of vascular Y1R regulation has been added to the Discussion (on **Page 23**) of our revised manuscript.

Meanwhile, it is worth noting that as a potent vasoconstrictor expressed in arterial smooth muscle, Y1R has been implicated in the pathology of hypertension (Tan et al., 2015). This vasodilation induced by the Y1R blockade could offer additional benefits to ameliorate hypertension, thereby improving overall cardiovascular function of obese or diabetic individuals who are at higher risk of cardiac complications. In line with this notion, a recent study has reported that the antagonism of Y1R confers protection against cardiac arrhythmia (Hoang et al., 2020). Moreover, we and others have also reported beneficial effects of peripheral Y1R antagonism in mouse models, including improving pancreatic islet beta cell function (Loh et al., 2017) and increasing bone mass (Sousa et al., 2012, Xie et al., 2020). Altogether, these positive findings from mouse studies suggest that systemic Y1R antagonism affords substantial benefits under various pathological conditions, further strengthening our conclusion. The possible effects of BIBO3304 on BAT temperature, cardiovascular function and blood pressure to have now been added to the Discussion (on **Page 23**) of our revised manuscript.

3. The ability of UCP-1 deletion to inhibit the effects of BIBO3304 on BW and BAT may not be specific to direct effects of Y1R on adipocytes. Loss of UCP-1 could prevent other more indirect effects as well, such as CNS or vascular actions. The authors should analyze Figure 8d and 8e as in Figure 1.

Our responses: Thank you for the comments. As UCP1 is exclusively expressed in BAT and browned WAT adipocytes, we believe that UCP1 deletion-induced abolishment of BIBO3304's effects on body weight and BAT is directly associated with Y1R effects on adipocytes. Having said that, it has been reported that UCP1 deletion in UCP1^{-/-} mice leads to more macromolecular alterations that is not limited to BAT (Kazak et al., 2017), and we agree with the reviewer that it is possible that loss of UCP1 could influence other effects of Y1R on non-adipose tissues such as blood vessels as discussed above, which could indirectly contribute to the overall changes in body weight and BAT function observed in BIBO3304-treated UCP1^{-/-} mice. We have now discussed this possibility (on **Page 21**) of our revised manuscript. As suggested, we have also re-analysed Figure 8d (Energy expenditure, EE) and 8e (RER) as in Figure 1. The EE data has now been normalized to the metabolically active tissues (MAT) (Even and Nadkarni, 2012, Zhang et al., 2018) to reflect the different contribution of lean mass and fat mass to the overall energy expenditure. For the details of normalization, please refer to **our responses below on Page 11**. The results have now been presented in **new Fig. 8f-g** (EE at light phase, dark phase and over 24-hour period) and **Fig. 8h-i** (RER at light phase, dark phase and over 24-hour period) in the revised manuscript. In addition to their actual body weight (**Fig 8a**) and whole-body fat mass (**new Fig. 8c**), we have also included body weight change (**new Fig. 8b**) and whole-body lean mass determined by DEXA (**new Fig. 8e**) to complete the phenotypic picture of UCP1^{-/-} mice in response to vehicle or BIBO3304 treatment.

4. A key experiment was determining if tamoxifen-induced loss of the Y1R in adult mice selectively in adipocytes mimics the effects of BIBO3304. However, the results are

confounded by the effects of tamoxifen alone to increase BW in mice eating chow or a HFD. It is difficult to understand how Y1R deletion prevents tamoxifen-induced weight gain, since the mechanism for this gain is unknown. It is also confounding that adipocyte Y1 deletion decreases this body weight gain in both chow- and HFD-fed mice, when BIBO3304 was effective only in HFD mice. This result requires further explanation and possibly experimentation.

Our responses: Thank you for the comments. Tamoxifen-induced weight gain has been reported before. This is most likely due to that tamoxifen is an estrogen receptor ligand, which elicits a persistent effect on adipocytes or preadipocytes via modulating proliferation, reported by Philip Scherer and his colleagues (Ye et al., 2015), whereas inhibiting Y1R in adipocytes suppresses MAPK-mediated proliferation (Yang et al., 2008). It is also possible that adipocyte Y1R deletion-induced elevation in thermogenesis outstrips the tamoxifen-induced adipogenesis, leading to a net outcome of no change in body weight. In addition, there is a possibility that tamoxifen-induced weight gain and Y1R deletion-induced normalization of body weight are mediated by independent pathways. Further studies are warranted to test these possibilities, but it is beyond the scope of this manuscript. We have provided further explanation to this phenomenon (on **Page 23-24**) in the revised manuscript.

Thank you also for pointing out that adipocyte-specific Y1R deletion decreased the body weight gain in both chow- and HFD-fed mice, when BIBO3304 was effective only in HFD mice. It is however, noteworthy that BIBO3304 treated chow-fed mice did display a trend towards a decrease in body weight gain albeit not reaching statistical significance (**Fig. 1g**). We have included this point into the Discussion (on **Page 24**) in the revised manuscript.

5. There are some inconsistencies that should be addressed or discussed: 1) BIBO3304 alters WAT_i versus WAT_i indices (e.g. weight, Y1R expression, size) variably. 2) A HFD/obesity increases UCP-1 in mice but decreases UCP-1 in humans.

Our responses: Thank you for the comments. As stated in our manuscript, activation of Y1R signaling *inhibits* the accumulation of cAMP, a direct activator of lipolysis and UCP1 expression (Park et al., 2014, Harms and Seale, 2013). Y1R expression is upregulated in WAT_i in obese mice and humans, suggesting a strong inverse relationship between Y1R expression in WAT_i and body weight as demonstrated in our manuscript (**Fig. 1d**). The treatment with BIBO3304, by removing the inhibitory action of Y1R on cAMP and promoting UCP1 expression, significantly reduces dissected WAT_i weights, which is attributable to markedly lower body weight gain especially under HFD feeding as shown in **Fig. 1h-i**. This reduction in WAT_i weight is associated with smaller adipocyte size in WAT_i that is due to increased thermogenic activity of beige adipocytes in WAT_i. To further confirm the WAT_i weight is not a confounding factor for Y1R expression, we have normalized the Y1R mRNA expression to WAT_i tissue weight used to extract total RNA from. As shown in **Reviewer Figure 3**, Y1R mRNA expression in HFD-fed WAT_i is still upregulated compared to chow-fed WAT_i when normalized to WAT_i weight.

With regards to the second question, it is well established that diet-induced adaptive thermogenesis is primarily mediated by UCP1 in mice (von Essen et al., 2017, Nedergaard and Cannon, 2013). There is a distinct expression profile of UCP1 in BAT and WAT_i under a positive energy balance induced by high fat feeding (Fromme and Klingenspor, 2011). In BAT, the prime organ for diet-induced thermogenesis, UCP1 expression is usually upregulated to

increase energy expenditure in order to limit weight gain and restore energy homeostasis in rodents and humans (Bachman et al., 2002, Rothwell and Stock, 1979, Bouchard et al., 1990); however, in WATi, the expression of UCP1 is downregulated with high fat feeding in mice and humans (Cinti, 2012, Fromme and Klingenspor, 2011, Himms-Hagen, 1979). More importantly, the abundance of UCP1-expressing beiging WATi is diminished in human obesity (Cinti, 2012). In agreement with this notion, we have harvested subcutaneous WAT, an equivalent fat pad of WATi in mice, from obese subjects and found UCP1 mRNA expression was markedly declined in obese individuals (**Fig. 10a**), which is consistent with decreased UCP1 expression in WATi of HFD-fed mice (Fromme and Klingenspor, 2011). Our results obtained in the current study are also consistent with these previous reports.

Reviewer Figure 3: Y1R mRNA expression in WATi normalized to the weights of WATi of C57Bl/6J mice fed a HFD for 7 weeks. (A) mRNA expression normalized to per mg WATi tissue weight; **(B)** mRNA expression normalized to WATi tissue weight and expressed relative to chow. Data are mean \pm s.e.m. of 3-4 mice. * $P < 0.05$ unpaired student t test (two tailed).

6. *The paper is descriptively written. It would benefit from a testable mechanistic hypothesis in the Introduction. In addition, the discussion is largely a restatement of the results, and the main conclusion is vague and more related to the potential use of BIBO3304 to counteract obesity. I recommend that the results and their discussion are built more like a story with mechanistic conclusions at each step. Instead, the discussion could include a cartoon that summarizes the findings and discusses in more detail the potential physiological and pathophysiological implications. For example, where does the NPY originate that binds BAT or WAT Y1R? How is this NPY regulated physiologically or pathophysiologicaly?*

Our responses: Thank you for your suggestions. We understand that we have employed a writing style that is different from the reviewer's preference. However, by using this writing style, we believe we have effectively conveyed our key messages; moreover, the other two reviewers have not raised any concerns regarding our writing style and so we believe there is no need for a major re-write. Having said this, we have taken the reviewer's comment on board and put forward a testable mechanistic hypothesis in the introduction (on **Page 4**), changed and expanded the discussion to address all the comments raised by the reviewer, and highlighted the potential physiological and pathophysiological implications of our findings (on **Page 20-21**). In addition, as suggested by the reviewer, we have included a schematic representing cartoon summarizing our major findings (**Reviewer Figure 4**).

Reviewer Figure 4: Schematic diagram summarizing the major findings of this study.

MINOR SPECIFIC COMMENTS.

1. Methodological concerns:

How was food intake measured?

Our response: Please refer to the response in “Point 1 in Major Specific Comments” above. Food intake measurement has been described extensively in our previous studies, and more details of methods (on **Page 28**) have been included in our revised manuscript.

How were mouse lumbar and BAT temperatures measured if the mice were unrestrained?

Our response: As described in our original manuscript, mouse lumbar back and BAT temperature was measured in unstrained freely-moving mice using non-invasive thermography following our well-established protocol that has been previously published (Shi et al., 2013, Zhang et al., 2014, Zhang et al., 2018, Farzi et al., 2018). Briefly, mice were single housed, fed either a chow or HFD, and had been acclimatized in their home cages for 3 days before measurement. One day prior to infrared imaging, all the testing mice had their interscapular and lumbar back regions shaved under light isoflurane anaesthesia to expose the skin of these areas for temperature measurement. A high-sensitivity infrared camera (ThermoCAM T640, FLIR, Danderyd, Sweden, sensitivity ± 0.04 °C) fixed on a tripod was placed 90 cm above the freely moving single-housed mouse to record the surface skin temperature of the mouse, and thermographic images (videos) were taken for 1 minute per cage. After each 1-minute measurement, the camera was moved to the next cage. Upon completion of the first round of measurement, a second round and third round were conducted. Temperature measurement was conducted once a day for 2 consecutive days at the same time of the day under ambient temperature conditions (22°C). The videos were analyzed with the ThermaCAM 7 Pro software

and the thermo frames that had mouse body naturally extended with both shaved skin regions vertical to camera lenses were extracted from the videos. The hottest pixels of the lumbar back, interscapular regions and tail (1cm from the base of tail) from each extracted frame were obtained simultaneously and used as indicators of body, BAT and tail temperature, respectively. The average temperature of lumbar back, interscapular BAT and tail was calculated. We have added more details of the temperature measurement and data analysis in the Methods (on **Page 30**) of our revised manuscript.

Should energy expenditure be normalized to body weight, as more fat would increase energy expenditure independently of metabolism or activity level.

Our response: Thank you for the comments. Energy expenditure shown in the original manuscript has been normalized by using lean mass as a covariate. Alternatively, energy expenditure (EE) can also be normalized to the metabolically active tissue (MAT) to reflect the primary contribution of lean mass but also the lesser contribution from adipose tissue to total energy expenditure. We have now normalized energy expenditure to metabolically active tissue (MAT) = Lean mass + Fat mass x 0.2 according to (Even and Nadkarni, 2012) (Zhang et al., 2018) (**Reviewer Figure 5**). This data is now presented as **new Fig. 2d-e**.

To be consistent, energy expenditure (EE) in HFD-fed UCP1^{-/-} mice fed a vehicle or BIBO3304 jelly in **Fig. 8** has also been normalized to MAT, and is presented now in **new Fig. 8f-g** in the revised manuscript. We have included a more detailed description of how EE and RER are measured and calculated as well as the method of normalization (on **Page 29**) in the revised manuscript.

Reviewer Figure 5. Energy expenditure normalized to metabolically active tissues (MAT) of chow-and HFD-fed C57BL/6J mice treated with BIBO3304 or vehicle containing jelly. Values are mean \pm s.e.m. of 6-10 mice. * $P < 0.05$ versus HFD control group determined by two-way ANOVA analysis.

Do differences in adipocyte size (or fat content) contribute to the PCR/Western results?

Our response: Differences in adipocyte size or fat content have been shown to potentially alter qPCR/western results of key molecules especially when comparing gene expressions between lean and obese rodents (Fischer et al., 2016, Nedergaard and Cannon, 2013, Fromme and Klingenspor, 2011). In obese or HFD-fed rodents, both brown adipose tissue (BAT) or white adipose tissue (WAT) are enlarged due to adipocyte hypertrophy, leading to increased tissue weights. However, because this increase is largely due to enlarged lipid droplet, the actual

protein density may be indeed decreased. Therefore, increased tissue mass under obese conditions with lower protein density results in similar total protein content of the tissue between lean and obese groups (Fischer et al., 2016). Therefore, it is suggested that the protein level of a gene of interest in a particular tissue can be normalized to the total amount of protein in the tissue (Fischer et al., 2016), as shown in our previous paper published in *Nature Communications* (Zhang et al., 2018). Moreover, in our study, the key comparison we wanted to make is the gene expression between control and BIBO3304 within the same diet. As shown in **Fig. 4d** (BAT) in our original manuscript, while BAT in HFD groups was heavier than that in chow groups as expected, BAT weight was not different between control and BIBO3304 treatment in either diet. Nevertheless, we went on to calculate the total protein content in BAT and WAT_i tissues that were used in western blotting. As shown in **Reviewer Figure 6**, there was no significant difference in total protein content per BAT or WAT_i between two treatments on either diet. We have then normalized the protein expression levels to the total protein content in BAT or WAT_i (**new Fig. 5b-c, new Fig. 6d-e and new Fig. 11a-d**) in our revised manuscript.

Reviewer Figure 6: Total amount of protein (mg) measured in (A) BAT and (B) WAT_i of chow- and HFD-fed mice treated with BIBO3304- or vehicle-containing jelly for 7 weeks. Values are mean \pm s.e.m. of 3-6 mice. Differences were determined by one-way ANOVA analysis.

What is the source of NPY in the cultures?

Our response: The source of NPY in the culture is likely derived by the adipocytes themselves as it has been shown that adipocytes synthesize and produce NPY in humans and mice (Sitticharoon et al., 2013, Yang et al., 2008), which is also shown in our hands (**Reviewer Figure 7A**) - NPY mRNA is expressed in human subcutaneous adipose tissue (SAT). To further confirm this, we have isolated stromal vascular fraction (SVF) containing pre-adipocytes in SAT of humans and induced them into fully differentiated mature adipocytes. RNA was extracted from SVF pre-adipocytes and mature adipocytes and reversely transcribed followed by qPCR. NPY mRNA expression was measured. As shown in **Reviewer Figure 7B** below, NPY is produced by mature adipocytes. We have now presented this data of NPY mRNA expression in SVF pre-adipocytes and mature adipocytes in **Supplementary Fig. 5** in our revised manuscript.

Reviewer Figure 7: NPY mRNA expression in human SAT (subcutaneous adipose tissue), SVF and differentiated mature adipocytes isolated from human SAT. (A) Ct values of NPY gene versus β -actin (housekeeping gene) determined by qPCR in human SAT. Values are Mean \pm s.e.m. of 6 samples per group. **(B)** NPY mRNA levels in SVF pre-adipocytes isolated from human SAT as well as differentiated mature adipocytes. Values are Mean \pm s.e.m. of 4 samples per group. Differences were determined by unpaired *t* test (two tailed).

2. Acknowledge that some of the data are confirmatory, e.g. p 6 and 8. PMID 23838112 and PMID 19918245.

Our response: We have acknowledged and included these two articles in our discussion in the revised manuscript.

3. Fig 10c. But the suppression of UCP-1 by NPY was not reversed by BIBO3304?

Our response: This is an interesting observation; we think the difference in UCP1 expression in response to the combination treatment could be a timing issue. It is possible that *ex vivo* induction of UCP1 mRNA expression in human SAT may require longer than 48-hour incubation time. We have made a statement regarding this in the text (on Page 17) in our revised manuscript.

References:

- BACHMAN, E. S., DHILLON, H., ZHANG, C. Y., CINTI, S., BIANCO, A. C., KOBILKA, B. K. & LOWELL, B. B. 2002. betaAR signaling required for diet-induced thermogenesis and obesity resistance. *Science*, 297, 843-5.
- BOUCHARD, C., TREMBLAY, A., DESPRES, J. P., NADEAU, A., LUPIEN, P. J., THERIAULT, G., DUSSAULT, J., MOORJANI, S., PINAULT, S. & FOURNIER, G. 1990. The response to long-term overfeeding in identical twins. *N Engl J Med*, 322, 1477-82.
- CHAVEZ, J. A., KNOTTS, T. A., WANG, L. P., LI, G., DOBROWSKY, R. T., FLORANT, G. L. & SUMMERS, S. A. 2003. A role for ceramide, but not diacylglycerol, in the antagonism of insulin signal transduction by saturated fatty acids. *J Biol Chem*, 278, 10297-303.
- CINTI, S. 2012. The adipose organ at a glance. *Dis Model Mech*, 5, 588-94.

- CONWAY, J. R. W., WARREN, S. C., HERRMANN, D., MURPHY, K. J., CAZET, A. S., VENNIN, C., SHEARER, R. F., KILLEN, M. J., MAGENAU, A., MELENEC, P., PINESE, M., NOBIS, M., ZARATZIAN, A., BOULGHOURJIAN, A., DA SILVA, A. M., DEL MONTE-NIETO, G., ADAM, A. S. A., HARVEY, R. P., HAIGH, J. J., WANG, Y., CROUCHER, D. R., SANSOM, O. J., PAJIC, M., CALDON, C. E., MORTON, J. P. & TIMPSON, P. 2018. Intravital Imaging to Monitor Therapeutic Response in Moving Hypoxic Regions Resistant to PI3K Pathway Targeting in Pancreatic Cancer. *Cell Rep*, 23, 3312-3326.
- CZECH, M. P. 2020. Mechanisms of insulin resistance related to white, beige, and brown adipocytes. *Mol Metab*, 34, 27-42.
- EVEN, P. C. & NADKARNI, N. A. 2012. Indirect calorimetry in laboratory mice and rats: principles, practical considerations, interpretation and perspectives. *Am J Physiol Regul Integr Comp Physiol*, 303, R459-76.
- FARZI, A., LAU, J., IP, C. K., QI, Y., SHI, Y. C., ZHANG, L., TASAN, R., SPERK, G. & HERZOG, H. 2018. Arcuate nucleus and lateral hypothalamic CART neurons in the mouse brain exert opposing effects on energy expenditure. *Elife*, 7.
- FISCHER, A. W., HOEFIG, C. S., ABREU-VIEIRA, G., DE JONG, J. M. A., PETROVIC, N., MITTAG, J., CANNON, B. & NEDERGAARD, J. 2016. Leptin Raises Defended Body Temperature without Activating Thermogenesis. *Cell Rep*, 14, 1621-1631.
- FROMME, T. & KLINGENSPOR, M. 2011. Uncoupling protein 1 expression and high-fat diets. *Am J Physiol Regul Integr Comp Physiol*, 300, R1-8.
- HARMS, M. & SEALE, P. 2013. Brown and beige fat: development, function and therapeutic potential. *Nat Med*, 19, 1252-63.
- HIMMS-HAGEN, J. 1979. Obesity may be due to a malfunctioning of brown fat. *Can Med Assoc J*, 121, 1361-4.
- HIRSCH, D. & ZUKOWSKA, Z. 2012. NPY and stress 30 years later: the peripheral view. *Cell Mol Neurobiol*, 32, 645-59.
- HOANG, J. D., SALAVATIAN, S., YAMAGUCHI, N., SWID, M. A. & VASEGHI, M. 2020. Cardiac sympathetic activation circumvents high-dose beta blocker therapy in part through release of neuropeptide Y. *JCI Insight*, 5.
- HUANG, X., LIU, G., GUO, J. & SU, Z. 2018. The PI3K/AKT pathway in obesity and type 2 diabetes. *Int J Biol Sci*, 14, 1483-1496.
- IP, C. K., ZHANG, L., FARZI, A., QI, Y., CLARKE, I., REED, F., SHI, Y. C., ENRIQUEZ, R., DAYAS, C., GRAHAM, B., BEGG, D., BRUNING, J. C., LEE, N. J., HERNANDEZ-SANCHEZ, D., GOPALASINGAM, G., KOLLER, J., TASAN, R., SPERK, G. & HERZOG, H. 2019. Amygdala NPY Circuits Promote the Development of Accelerated Obesity under Chronic Stress Conditions. *Cell Metab*, 30, 111-128 e6.
- KAZAK, L., CHOUCANI, E. T., STAVROVSKAYA, I. G., LU, G. Z., JEDRYCHOWSKI, M. P., EGAN, D. F., KUMARI, M., KONG, X., ERICKSON, B. K., SZPYT, J., ROSEN, E. D., MURPHY, M. P., KRISTAL, B. S., GYGI, S. P. & SPIEGELMAN, B. M. 2017. UCP1 deficiency causes brown fat respiratory chain depletion and sensitizes mitochondria to calcium overload-induced dysfunction. *Proc Natl Acad Sci U S A*, 114, 7981-7986.
- KOMATSU, N., AOKI, K., YAMADA, M., YUKINAGA, H., FUJITA, Y., KAMIOKA, Y. & MATSUDA, M. 2011. Development of an optimized backbone of FRET biosensors for kinases and GTPases. *Mol Biol Cell*, 22, 4647-56.
- LOH, K., SHI, Y. C., WALTERS, S., BENSELLAM, M., LEE, K., DEZAKI, K., NAKATA, M., IP, C. K., CHAN, J. Y., GURZOV, E. N., THOMAS, H. E., WAIBEL, M., CANTLEY, J., KAY, T. W.,

- YADA, T., LAYBUTT, D. R., GREY, S. T. & HERZOG, H. 2017. Inhibition of Y1 receptor signaling improves islet transplant outcome. *Nat Commun*, 8, 490.
- NEDERGAARD, J. & CANNON, B. 2013. UCP1 mRNA does not produce heat. *Biochim Biophys Acta*, 1831, 943-9.
- NGUYEN, A. D., MITCHELL, N. F., LIN, S., MACIA, L., YULYANINGSIH, E., BALDOCK, P. A., ENRIQUEZ, R. F., ZHANG, L., SHI, Y. C., ZOLOTUKHIN, S., HERZOG, H. & SAINSBURY, A. 2012. Y1 and Y5 receptors are both required for the regulation of food intake and energy homeostasis in mice. *PLoS One*, 7, e40191.
- PARK, S., FUJISHITA, C., KOMATSU, T., KIM, S. E., CHIBA, T., MORI, R. & SHIMOKAWA, I. 2014. NPY antagonism reduces adiposity and attenuates age-related imbalance of adipose tissue metabolism. *FASEB J*, 28, 5337-48.
- ROTHWELL, N. J. & STOCK, M. J. 1979. A role for brown adipose tissue in diet-induced thermogenesis. *Nature*, 281, 31-5.
- SCHINDELIN, J., ARGANDA-CARRERAS, I., FRISE, E., KAYNIG, V., LONGAIR, M., PIETZSCH, T., PREIBISCH, S., RUEDEN, C., SAALFELD, S., SCHMID, B., TINEVEZ, J. Y., WHITE, D. J., HARTENSTEIN, V., ELICEIRI, K., TOMANCAK, P. & CARDONA, A. 2012. Fiji: an open-source platform for biological-image analysis. *Nat Methods*, 9, 676-82.
- SHI, Y. C., LAU, J., LIN, Z., ZHANG, H., ZHAI, L., SPERK, G., HEILBRONN, R., MIETZSCH, M., WEGER, S., HUANG, X. F., ENRIQUEZ, R. F., BALDOCK, P. A., ZHANG, L., SAINSBURY, A., HERZOG, H. & LIN, S. 2013. Arcuate NPY controls sympathetic output and BAT function via a relay of tyrosine hydroxylase neurons in the PVN. *Cell Metab*, 17, 236-48.
- SITTICHAROON, C., CHATREE, S. & CHURINTARAPHAN, M. 2013. Expressions of neuropeptide Y and Y1 receptor in subcutaneous and visceral fat tissues in normal weight and obese humans and their correlations with clinical parameters and peripheral metabolic factors. *Regul Pept*, 185, 65-72.
- SOUSA, D. M., BALDOCK, P. A., ENRIQUEZ, R. F., ZHANG, L., SAINSBURY, A., LAMGHARI, M. & HERZOG, H. 2012. Neuropeptide Y Y1 receptor antagonism increases bone mass in mice. *Bone*, 51, 8-16.
- TAN, S. X., FISHER-WELLMAN, K. H., FAZAKERLEY, D. J., NG, Y., PANT, H., LI, J., MEOLI, C. C., COSTER, A. C., STOCKLI, J. & JAMES, D. E. 2015. Selective insulin resistance in adipocytes. *J Biol Chem*, 290, 11337-48.
- TERUEL, T., HERNANDEZ, R. & LORENZO, M. 2001. Ceramide mediates insulin resistance by tumor necrosis factor-alpha in brown adipocytes by maintaining Akt in an inactive dephosphorylated state. *Diabetes*, 50, 2563-71.
- VON ESSEN, G., LINDSUND, E., CANNON, B. & NEDERGAARD, J. 2017. Adaptive facultative diet-induced thermogenesis in wild-type but not in UCP1-ablated mice. *Am J Physiol Endocrinol Metab*, 313, E515-E527.
- WIELAND, H. A., ENGEL, W., EBERLEIN, W., RUDOLF, K. & DOODS, H. N. 1998. Subtype selectivity of the novel nonpeptide neuropeptide Y Y1 receptor antagonist BIBO 3304 and its effect on feeding in rodents. *Br J Pharmacol*, 125, 549-55.
- WINZELL, M. S. & AHREN, B. 2004. The high-fat diet-fed mouse: a model for studying mechanisms and treatment of impaired glucose tolerance and type 2 diabetes. *Diabetes*, 53 Suppl 3, S215-9.
- XIE, W., LI, F., HAN, Y., QIN, Y., WANG, Y., CHI, X., XIAO, J. & LI, Z. 2020. Neuropeptide Y1 receptor antagonist promotes osteoporosis and microdamage repair and enhances

- osteogenic differentiation of bone marrow stem cells via cAMP/PKA/CREB pathway. *Aging (Albany NY)*, 12, 8120-8136.
- YANG, K., GUAN, H., ARANY, E., HILL, D. J. & CAO, X. 2008. Neuropeptide Y is produced in visceral adipose tissue and promotes proliferation of adipocyte precursor cells via the Y1 receptor. *FASEB J*, 22, 2452-64.
- YE, R., WANG, Q. A., TAO, C., VISHVANATH, L., SHAO, M., MCDONALD, J. G., GUPTA, R. K. & SCHERER, P. E. 2015. Impact of tamoxifen on adipocyte lineage tracing: Inducer of adipogenesis and prolonged nuclear translocation of Cre recombinase. *Mol Metab*, 4, 771-8.
- ZHANG, L., IP, C. K., LEE, I. J., QI, Y., REED, F., KARL, T., LOW, J. K., ENRIQUEZ, R. F., LEE, N. J., BALDOCK, P. A. & HERZOG, H. 2018. Diet-induced adaptive thermogenesis requires neuropeptide FF receptor-2 signalling. *Nat Commun*, 9, 4722.
- ZHANG, L., LEE, I. C., ENRIQUEZ, R. F., LAU, J., VAHATALO, L. H., BALDOCK, P. A., SAVONTAUS, E. & HERZOG, H. 2014. Stress- and diet-induced fat gain is controlled by NPY in catecholaminergic neurons. *Mol Metab*, 3, 581-91.

Reviewer comments, second round –

Reviewer #2 (Remarks to the Author):

In the revised manuscript, the authors added more detailed descriptions of the two-photon lifetime imaging experiment using the Akt-FRET biosensor by referring the previous studies. In addition, the authors showed the clear difference of the lifetime imaging in brown adipose tissue with or without treatment of a mTOR1/2 inhibitor AZD2014, which validates the specificity of the biosensor in this study. The authors also gave a comment on the non-significant increase in Akt activity at the later time period in the original Figure 11f. These results and explanations sufficiently addressed my concerns and more strongly support the conclusion. Therefore, I recommend that this manuscript would be acceptable for publication in Nature Communications.

Reviewer #3 (Remarks to the Author):

The authors were quite responsive to comments and concerns. My only new comments are to check the revised (purple) segments for occasional typographical errors and awkward phrasing. Also in Figure 11, define bar colors either in the figure or legend.

Congratulations on a very novel and complete set of experiments and results.

REVIEWERS' COMMENTS

Reviewer #2 (Remarks to the Author):

In the revised manuscript, the authors added more detailed descriptions of the two-photon lifetime imaging experiment using the Akt-FRET biosensor by referring the previous studies. In addition, the authors showed the clear difference of the lifetime imaging in brown adipose tissue with or without treatment of a mTOR1/2 inhibitor AZD2014, which validates the specificity of the biosensor in this study. The authors also gave a comment on the non-significant increase in Akt activity at the later time period in the original Figure 11f. These results and explanations sufficiently addressed my concerns and more strongly support the conclusion. Therefore, I recommend that this manuscript would be acceptable for publication in Nature Communications.

Our response: We appreciate your positive comments on our revised manuscript.

Reviewer #3 (Remarks to the Author):

The authors were quite responsive to comments and concerns. My only new comments are to check the revised (purple) segments for occasional typographical errors and awkward phrasing. Also in Figure 11, define bar colors either in the figure or legend.

Congratulations on a very novel and complete set of experiments and results.

Our response: Thank you for your comments. We have checked the revised (purple) segments and corrected typographic errors and rephrased the sentences in page 2, page 4, page 16, page 19-20, page 23-24 and page 29 in our updated manuscript.

We have also defined bar colors in the legend of Figure 11 (now Fig.10) in our revised manuscript.